# Graph neural fields: A framework for spatiotemporal dynamical models on the human connectome

**Marco Aqil**[1¤]*, **Selen Atasoy**[2,3], **Morten L. Kringelbach**[2,3], **Rikkert Hindriks**[1]

**1** Department of Mathematics, Vrije Universiteit, Amsterdam, The Netherlands, **2** Centre for Eudaimonia and Human Flourishing, University of Oxford, Oxford, United Kingdom, **3** Center for Music in the Brain, University of Aarhus, Aarhus, Denmark

¤ Current address: Spinoza Centre for Neuroimaging, Amsterdam, The Netherlands
* m.aqil@spinozacentre.nl

**Data Availability Statement:** Data used in this work were made available by the Human Connectome Project (HCP), WU-Minn Consortium. We use fMRI, MRI and DTI data of an individual

## Abstract

Tools from the field of graph signal processing, in particular the graph Laplacian operator, have recently been successfully applied to the investigation of structure-function relationships in the human brain. The eigenvectors of the human connectome graph Laplacian, dubbed "connectome harmonics", have been shown to relate to the functionally relevant resting-state networks. Whole-brain modelling of brain activity combines structural connectivity with local dynamical models to provide insight into the large-scale functional organization of the human brain. In this study, we employ the graph Laplacian and its properties to define and implement a large class of neural activity models directly on the human connectome. These models, consisting of systems of stochastic integrodifferential equations on graphs, are dubbed *graph neural fields*, in analogy with the well-established continuous neural fields. We obtain analytic predictions for harmonic and temporal power spectra, as well as functional connectivity and coherence matrices, of graph neural fields, with a technique dubbed CHAOSS (shorthand for *Connectome-Harmonic Analysis Of Spatiotemporal Spectra*). Combining graph neural fields with appropriate observation models allows for estimating model parameters from experimental data as obtained from electroencephalography (EEG), magnetoencephalography (MEG), or functional magnetic resonance imaging (fMRI). As an example application, we study a stochastic Wilson-Cowan graph neural field model on a high-resolution connectome graph constructed from diffusion tensor imaging (DTI) and structural MRI data. We show that the model equilibrium fluctuations can reproduce the empirically observed harmonic power spectrum of resting-state fMRI data, and predict its functional connectivity, with a high level of detail. Graph neural fields natively allow the inclusion of important features of cortical anatomy and fast computations of observable quantities for comparison with multimodal empirical data. They thus appear particularly suitable for modelling whole-brain activity at mesoscopic scales, and opening new potential avenues for connectome-graph-based investigations of structure-function relationships.

subject (#100307), available from https://db.humanconnectome.org/data/projects/HCP_1200. All code used for analysis and simulations is available for use and review at https://github.com/marcoaqil/Graph-Stochastic-Wilson-Cowan-Model. Together, these two links contain all the data and all the code used for this work.

**Funding:** MLK and SA are supported by the ERC Consolidator Grant: CARE-GIVING (n. 615539), Center for Music in the Brain, funded by the Danish National Research Foundation (DNRF117), and Centre for Eudaimonia and Human Flourishing funded by the Pettit and Carlsberg Foundations. RH is supported by the NWO-Wiskundeclusters grant nr. 613.009.105. The funders had no role in study design, data collection and analysis, decision to publish, or preparation of the manuscript.

**Competing interests:** The authors have declared that no competing interests exist.

## Author summary

The human brain can be seen as an interconnected network of many thousands neuronal "populations"; in turn, each population contains thousands of neurons, and each is connected both to its neighbors on the cortex, and crucially also to distant populations thanks to long-range white matter fibers. This extremely complex network, unique to each of us, is known as the "human connectome graph". In this work, we develop a novel approach to investigate how the neural activity that is necessary for our life and experience of the world arises from an individual human connectome graph. For the first time, we implement a mathematical model of neuronal activity directly on a high-resolution connectome graph, and show that it can reproduce the spatial patterns of activity observed in the real brain with magnetic resonance imaging. This new kind of model, made of equations implemented directly on connectome graphs, could help us better understand how brain function is shaped by computational principles and anatomy, but also how it is affected by pathology and lesions.

## Introduction

The spatiotemporal dynamics of human resting-state brain activity is organized in functionally relevant ways, with perhaps the best-known example being the "resting-state networks" [1]. How the repertoire of resting-state brain activity arises from the underlying anatomical structure, i.e. the connectome, is a highly non-trivial question: it has been shown that structural connections imply functional ones, but that the converse is not necessarily true [2]; furthermore, specific discordant attributes of structural and functional connectivity have been found by network analyses [3, 4]. Research on structure-function questions can be broadly divided into data-driven (analysis), theory-driven (modelling), and combinations thereof. In this work, we combine techniques from graph signal processing (analysis) and neural field equations (modelling) to outline a promising new approach for the investigation of whole-brain structure-function relationships.

A recent trend of particular interest in neuroimaging data analysis is the application of methods from the field of graph signal processing [5–10]. In these applications, anatomical information obtained from DTI and structural MRI is used to construct the *connectome graph* [11], and combined with functional imaging data such as BOLD-fMRI or EEG/MEG to investigate structure-function relationships in the human brain (see [12, 13] for reviews). The workhorse of graph signal processing analysis is the *graph Laplacian operator*, or simply graph Laplacian. Originally formulated as the graph-equivalent of the Laplace-Beltrami operator for Riemannian manifolds [14, 15], the graph Laplacian is now established as a valuable tool in its own right [12]. The eigenvectors of the graph Laplacian provide a generalization of the Fourier transform to graphs, and therefore also a complete orthogonal basis for functions on the graph. In the context of the human connectome graph, the eigenvectors of the graph Laplacian are referred to as *connectome harmonics*, by analogy with the harmonic eigenfunctions of the Laplace-Beltrami operator. Of relevance to the current work, several connectome harmonics have been shown to be related to specific resting-state networks [11]. More recent studies have provided additional evidence for this claim [16, 17], and others used a similar approach to explain how distinct electrophysiological resting-state networks emerge from the structural connectome graph [18]. Furthermore in [11], for the first time to the best of our knowledge, a model of neural activity making use of the graph

Laplacian was implemented, and used to suggest the role of Excitatory-Inhibitory dynamics as possible underlying mechanism for the self-organization of resting-state activity patterns. In other very recent work [19, 20] techniques based on the graph Laplacian were employed to model EEG and MEG oscillations. Considering these developments, the combination of neural activity modelling and graph signal processing techniques appears as a promising direction for further inquiry.

Whole-brain models are models of neural activity that are defined on the entire cortex and possibly on subcortical structures. This is generally achieved either by parcellating the cortex into a network of a few dozens of macroscopic, coupled regions of interest (ROIs), or by approximating the cortex as a bidimensional manifold, and studying continuous integrodifferential equations in a flat or spherical geometry (see [21] for a review). In this study, relying on graph signal processing methods such as the graph Laplacian and graph filtering [7, 9], we show how to define and implement a large class of whole-brain models of neural activity on arbitrary metric graphs (that is, graphs equipped with a distance metric), and in particular on an unparcellated, mesoscopic-resolution human connectome. These models consist of systems of stochastic integrodifferential equations on graphs, and we refer to them as *graph neural fields* by analogy with their continuous counterparts. We obtain analytic expressions for harmonic and temporal power spectra, as well as functional connectivity and coherence matrices of graph neural fields, with a technique dubbed CHAOSS (shorthand for *Connectome-Harmonic Analysis Of Spatiotemporal Spectra*). When combined with appropriate observation models, graph neural fields can be fitted to and compared with functional data obtained from different imaging modalities such as EEG/MEG, fMRI, and positron emission tomography (PET). Graph neural fields can take into account many physical properties of the cortex, and provide a computationally efficient and versatile modelling framework that is tailored for connectome-graph-based structure-function investigations, particularly suitable for modelling whole-brain activity on mesoscopic scales. Graph neural fields present immediate application in the investigation of the relationship between individual anatomy, pathology, lesions, neuropharmacological alterations, with functional brain activity; and furthermore provide a model-based approach to test novel graph signal processing neuroimaging hypotheses. While here we focus on the human connectome as a prime application for graph neural fields, the mathematical framework can also be used to implement and analyze single-neuron models directly on the connectome graphs of simple organisms, such as *C. Elegans*, whose full neuronal pathways have been experimentally mapped [22].

In Results, we implement, analyze, and numerically simulate a stochastic Wilson-Cowan graph neural field model, first on a one-dimensional graph with 1000 vertices, and then on a single-subject multimodal connectome consisting of approximately 18000 cortical vertices and 15000 white matter tracts. The simplified context of a one-dimensional graph is useful to illustrate the effect of graph properties, such as distance weighting and non-local edges, on model dynamics; moving on to a real-world application, we show that the model implemented on the full connectome can reproduce the experimentally observed harmonic power spectrum of resting-state fMRI data, and predict the fMRI functional connectivity matrix with a high level of detail. In Methods, we describe the general framework of graph neural fields, and show how to derive analytic expressions for harmonic and temporal power spectra, as well as coherence and functional connectivity matrices (CHAOSS). Methodological generalizations, full linear stability analysis of the Wilson-Cowan graph neural field model, and an implementation of the damped-wave equation on the human connectome graph, are provided in S1–S4 Appendices.

## Results

### Stochastic Wilson-Cowan equations on graphs

The Wilson-Cowan model [23] is a widely used and successful model of cortical dynamics. In this section we show how to use the framework of graph neural fields to implement the stochastic Wilson-Cowan equations on an arbitrary graphs equipped with a distance metric, and how to compute spatiotemporal observables (CHAOSS). We then illustrate the effects of distance-weighting and non-local graph edges on model dynamics in the simplified context of a one-dimensional graph, before moving on to a real-world application with fMRI data.

In continuous space, the stochastic Wilson-Cowan model is described by the following system of integrodifferential equations:

$$\tau_E \frac{\partial E}{\partial t} = -d_E E + S[\alpha_{EE}(K_{EE} \otimes E) - \alpha_{IE}(K_{IE} \otimes I) + P] + \sigma \xi_E, \tag{1}$$

$$\tau_I \frac{\partial I}{\partial t} = -d_I I + S[\alpha_{EI}(K_{EI} \otimes E) - \alpha_{II}(K_{II} \otimes I) + Q] + \sigma \xi_I, \tag{2}$$

where $\otimes$ denotes a convolution integral, and we have omitted for brevity the spatiotemporal dependency of $E(x, t)$, $I(x, t)$, $\xi_E(x, t)$ and $\xi_I(x, t)$. This model posits the existence of two neuronal populations (Excitatory and Inhibitory) at each location in space. The fractions of active neurons in each population ($E$, $I$) evolve according to a spontaneous decay with rate $d_E$ and $d_I$, a sigmoid-mediated activation term containing the four combinations of population interactions (*E-E, I-E, E-I, I-I*) as well as the subcortical input terms $P$ and $Q$, stochastic noise realizations $\xi_E$ and $\xi_I$ of intensity $\sigma$, and with the timescale parameters $\tau_E$ and $\tau_I$. The propagation of activity and interaction among neuronal populations is modeled by spatial convolution integrals with four, potentially different, kernels ($K_{EE}$, $K_{IE}$, $K_{EI}$, $K_{II}$). For arbitrary spatially symmetric kernels, convolution integrals can be formulated on graphs as linear matrix-vector products (Eq (28)). Table 1 summarizes the meaning of symbols in the Wilson-Cowan equations.

**Table 1. Meaning of symbols in the Wilson-Cowan equations.**

| Symbol | Meaning |
|---|---|
| $E$ | Fraction of active Excitatory neurons in local populations. |
| $I$ | Fraction of active Inhibitory neurons in local populations. |
| $\tau_E$, $\tau_I$ | Excitatory/Inhibitory timescale parameters. |
| $d_E$, $d_I$ | Excitatory/Inhibitory spontaneous activity decay rates. |
| $S[x]$ | Sigmoid function $1/(1+e^{-x})$. |
| $\alpha_{EE}$, $\alpha_{IE}$, $\alpha_{EI}$, $\alpha_{II}$ | Strength of connectivity between pairs of neuronal populations. |
| $K_{EE}$, $K_{IE}$, $K_{EI}$, $K_{II}$ | Convolution kernels in continuous space, corresponding filters on graphs. |
| $\sigma_{EE}$, $\sigma_{IE}$, $\sigma_{EI}$, $\sigma_{II}$ | Standard deviation of Gaussian kernels/filters. |
| $P$ | Subcortical input to Excitatory populations. |
| $Q$ | Subcortical input to Inhibitory populations. |
| $\sigma$ | Noise amplitude. |
| $\xi_E$, $\xi_I$ | Noise realizations. |

The Wilson-Cowan equations model the spatiotemporal dynamics of interactions among Excitatory and Inhibitory neuronal populations. Note that each of the four possible pairs of population interactions is described by a distinct kernel/filter. Here, we use four Gaussian kernels of different sizes.

**Table 2. Spatial convolution kernels in Euclidean, Fourier, and graph domains.**

| Kernel | Euclidean domain | Fourier domain | $\hat{K}_g$ |
|:---:|:---:|:---:|:---:|
| Gaussian | $e^{-x^2/2\sigma^2}$ | $e^{-\sigma^2 k^2/2}$ | $e^{\sigma^2 \lambda_k/2}$ |
| Exponential | $e^{-\alpha|x|}$ | $\frac{1}{\alpha^2+k^2}$ | $\frac{1}{\alpha^2-\lambda_k}$ |
| Mexican hat | $(1-(x/\sigma)^2)e^{-x^2/2\sigma^2}$ | $k^2 e^{-\sigma^2 k^2/2}$ | $-\lambda_k e^{\sigma^2 \lambda_k/2}$ |
| Rectangular | $\mathrm{rect}(ax)$ | $\mathrm{sinc}\left(\frac{k}{2\pi a}\right)$ | $\mathrm{sinc}\left(\frac{\sqrt{-\lambda_k}}{2\pi a}\right)$ |
| Triangular | $\mathrm{tri}(ax)$ | $\mathrm{sinc}^2\left(\frac{k}{2\pi a}\right)$ | $\mathrm{sinc}^2\left(\frac{\sqrt{-\lambda_k}}{2\pi a}\right)$ |

This table provides examples of commonly used continuous convolution kernels and their graph-domain equivalents. In short, substituting $k^2$ with $-\lambda_k$ in the Fourier transform of a continuous kernel provides its graph-domain translation. The choice of kernel and the value of kernel parameters (for example the size $\sigma$ of a Gaussian kernel) have a significant influence on model dynamics. Normalization factors are omitted. The function sinc is defined as $\mathrm{sinc}(x) = \sin(\pi x)/(\pi x)$.

Thus, the stochastic Wilson-Cowan graph neural field model can be formulated as:

$$\tau_E \frac{dE}{dt} = -d_E E + S[\alpha_{EE} K_{EE} E - \alpha_{IE} K_{IE} I + P] + \sigma \xi_E, \tag{3}$$

$$\tau_I \frac{dI}{dt} = -d_I I + S[\alpha_{EI} K_{EI} E - \alpha_{II} K_{II} I + Q] + \sigma \xi_I, \tag{4}$$

where $E$, $I$, $\xi_E$ and $\xi_I$ are functions on the graph, i.e. vectors of size $n$, where $n$ is the number of vertices in the graph. The convolution integrals are implemented via the graph-filters $K_{**}$, which are matrices of size $(n, n)$. In particular, for the case of Gaussian kernels, the filters are given by (Table 2):

$$K_{EE} = U e^{\sigma_{EE}^2 \Lambda/2} U^T, \quad K_{IE} = U e^{\sigma_{IE}^2 \Lambda/2} U^T, \tag{5}$$

$$K_{EI} = U e^{\sigma_{EI}^2 \Lambda/2} U^T, \quad K_{II} = U e^{\sigma_{II}^2 \Lambda/2} U^T, \tag{6}$$

where $\Delta = U^T \Lambda U$ denotes the distance-weighted graph Laplacian and its diagonalization (Eqs (22 and 23)). Note that each kernel has a different size parameter $\sigma_{**}$, effectively allowing different spatial ranges for Excitatory and Inhibitory interactions, without requiring a Mexican-hat kernel. Importantly, the inclusion of a stochastic noise term in the model formulation allows for characterization of resting-state activity as noise-induced fluctuations about a stable steady-state $(E^*, I^*)$ [24].

## Wilson-Cowan model CHAOSS

Having defined the Wilson-Cowan graph neural field equations, we wish to apply the *Connectome-Harmonic Analysis Of Spatiotemporal Spectra* to characterize the dynamics of resting-state fluctuations in neural activity. CHAOSS predictions, combined with a suitable observation model, can then be compared with empirical neuroimaging data, for example EEG, MEG, or fMRI. To do this, we obtain the linearized Wilson-Cowan equations for the evolution of a

perturbation about a steady state:

$$\tau_E \frac{dE}{dt} = -d_E E + a\alpha_{EE} K_{EE} E - a\alpha_{IE} K_{IE} I + \sigma\xi_E, \tag{7}$$

$$\tau_I \frac{dI}{dt} = -d_I I + b\alpha_{EI} K_{EI} E - b\alpha_{II} K_{II} I + \sigma\xi_I, \tag{8}$$

where the scalar, steady-state-dependent parameters $a$ and $b$ are:

$$a = d_E E^*(1 - d_E E^*), \qquad b = d_I I^*(1 - d_I I^*). \tag{9}$$

Derivation of the linearized equations and their full linear stability analysis can be found in S4 Appendix. In the graph Fourier domain, Eqs (7 and 8) are diagonalized and can be recast in the standard form:

$$\frac{d\hat{u}_k}{dt} = J_k \hat{u}_k + \sqrt{B}\hat{\xi}_k, \tag{10}$$

where the vector $u$ contains the concatenation of population activities on the graph $E$ and $I$, $\xi$ contains the concatenation of noise realizations $\xi_E$ and $\xi_I$. The hat notation $\hat{u}$ indicates the graph Fourier transform, and $k = 1, \ldots, n$ indexes the graph Laplacian eigenmodes. For the Wilson-Cowan model with Gaussian kernels, the Jacobian of the $k^{th}$ eigenmode is:

$$J_k = \begin{bmatrix} -\frac{d_E}{\tau_E} + \frac{a}{\tau_E}\alpha_{EE}e^{\sigma_{EE}^2\lambda_k/2} & -\frac{a}{\tau_E}\alpha_{IE}e^{\sigma_{IE}^2\lambda_k/2} \\ \\ \frac{b}{\tau_I}\alpha_{EI}e^{\sigma_{EI}^2\lambda_k/2} & -\frac{b}{\tau_I}\alpha_{II}e^{\sigma_{II}^2\lambda/2} - \frac{d_I}{\tau_I} \end{bmatrix}, \tag{11}$$

where $\lambda_k$ is the $k^{th}$ graph Laplacian eigenvalue, and:

$$B = \begin{bmatrix} \sigma^2/\tau_E^2 & 0 \\ 0 & \sigma^2/\tau_I^2 \end{bmatrix}. \tag{12}$$

In terms of the elements of the matrices $J_k$ and $B$, the two-dimensional harmonic-temporal power spectrum of the Excitatory neuronal population activity is (Eq (41)):

$$[S_k(\omega)]_E = \frac{[B]_{00}([J_k]_{11}^2 + \omega^2) + [J_k]_{01}^2[B]_{11}}{([J_k]_{00}[J_k]_{11} - [J_k]_{01}[J_k]_{10} - \omega^2)^2 + \omega^2([J_k]_{00} + [J_k]_{11})^2}. \tag{13}$$

The double-digits numerical subscripts refer to the row-column element of the respective matrix. Eq (13) describes the power of Excitatory activity, in the $k^{th}$ eigenmode, at temporal frequency $\omega$. It can be used to compute the separate harmonic and temporal power spectra, as well as the functional connectivity and coherence matrices of the model. Equivalent formulas for the Inhibitory population can also be derived.

By integrating $[S_k(\omega)]_E$ over all temporal frequencies, an explicit expression for the harmonic power spectrum of Excitatory activity can be obtained:

$$H_E(k) = \frac{[B]_{00}([J_k]_{00}[J_k]_{11} - [J_k]_{01}[J_k]_{10}) + [J_k]_{11}^2[B]_{00} + [J_k]_{01}^2[B]_{11}}{2([J_k]_{01}[J_k]_{10} - [J_k]_{00}[J_k]_{11})([J_k]_{00} + [J_k]_{11})}. \tag{14}$$

Similarly, the temporal power spectrum can be obtained by summing $[S_k(\omega)]_E$ over all harmonic eigenmodes (Eq (42)). Eqs (13) and (14) represent a *general* result that does not only

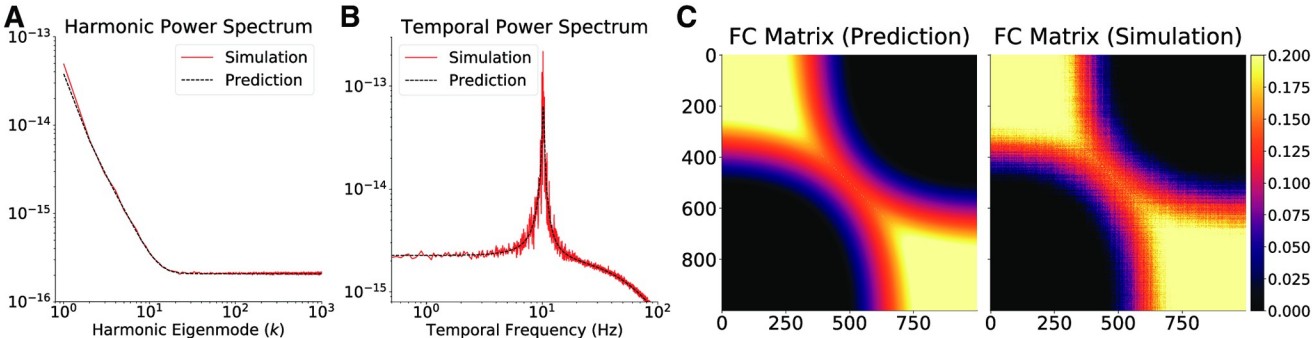

**Fig 1. Effects of distance-weighting on graph neural field dynamics.** (A) The harmonic and (B) temporal power spectra of Excitatory activity equilibrium fluctuations in the one-dimensional graph for vertex spacing $h = 10^{-4}$ $m$. A larger vertex spacing, for example $h = 2 \cdot 10^{-4}$ $m$, renders the steady state unstable. The dashed black lines correspond to the theoretical prediction and the red lines are obtained through numerical simulations. (C) The Excitatory activity functional connectivity obtained by analytic predictions and numerical simulations.

apply to the Wilson-Cowan model. In fact, these equations describe the power spectra of stochastic equilibrium fluctuations for the first population of any graph neural field model with two interacting populations and a first-order temporal differential operator. The specific shape of the power spectra will depend on the model formulation and on its parameters.

**Effects of distance-weighting and non-local connectivity.** Distance-weighting of graph edges, presence of non-local connectivity, and changes in model parameter values can have significant effects on dynamics of graph neural fields. To demonstrate this, we implement the stochastic Wilson-Cowan model in the simplified context of a one-dimensional graph with 1000 vertices. Numerical simulations were carried out with a time-step $\delta t = 5 \cdot 10^{-5}$ seconds, for a total time of 20 seconds of simulated activity ($4 \cdot 10^5$ time-steps). The parameter set for the results shown in Figs 1–3 is reported in S1 Table. In Fig 4, the value of $\sigma_{IE}$ is increased by a factor of 20, with everything else unchanged, as an illustrative example of the influence of kernel parameters on model dynamics.

To show the effects of distance-weighting in graph neural fields, we note how, for the parameter set of S1 Table, increasing the distance between vertices leads to the emergence of an oscillatory resonance that eventually destabilizes the steady state and gives way to limit-cycle activity. Keeping the number of vertices constant, increasing the vertex spacing $h$ alters

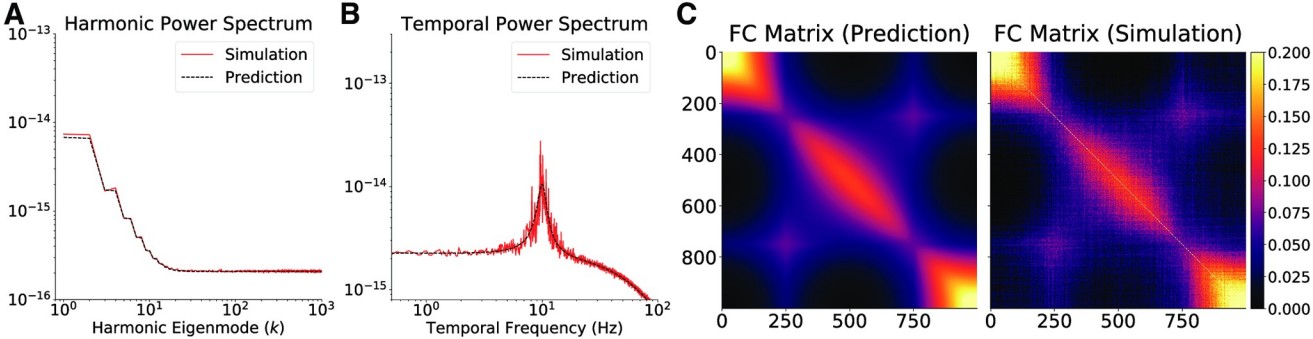

**Fig 2. Suppression of oscillatory resonance by non-local connectivity.** (A) The harmonic and (B) temporal power spectra of Excitatory activity equilibrium fluctuations in the one-dimensional graph for vertex spacing $h = 10^{-4}$ $m$, after the addition of a non-local edge between vertices 250 and 750. The dashed black lines correspond to the theoretical prediction and the red lines are obtained through numerical simulations. (C) The Excitatory activity functional connectivity obtained by analytic predictions and numerical simulations. Compare with Fig 1 to note the visible suppression of oscillatory resonance in the temporal power spectrum, and the change in functional connectivity engendered by a single non-local edge.

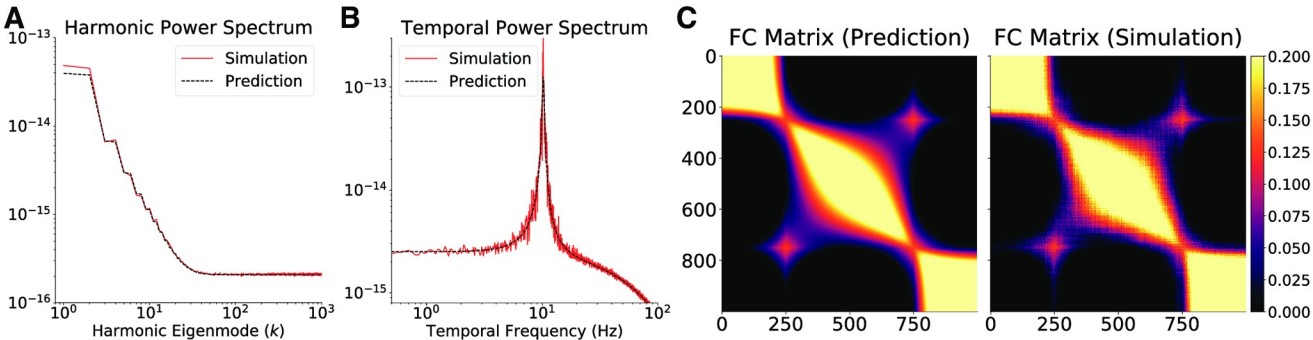

**Fig 3. Abortion of pathological oscillations by non-local connectivity.** (A) The harmonic and (B) temporal power spectra of Excitatory activity equilibrium fluctuations in the one-dimensional graph for vertex spacing $h = 2 \cdot 10^{-4}\ m$, after the addition of a non-local edge between vertices 250 and 750. Without the addition, the model dynamics is placed in an unstable limit-cycle regime. The dashed black lines correspond to the theoretical prediction and the red lines are obtained through numerical simulations. (C) The Excitatory activity functional connectivity obtained by analytic predictions and numerical simulations.

the stability of the steady state from broadband activity ($h = 10^{-5}$m), to oscillatory resonance ($h = 10^{-4}$m), to oscillatory instability ($h = 2 \cdot 10^{-4}$m). This result demonstrates that the dynamics of graph neural fields are dependent on the metric properties of the graph, and hence indicate the necessity of employing the distance-weighted graph Laplacian in the context of graph neural fields modelling. The combinatorial (binary) graph Laplacian captures the topology, but not the geometry, of the graph, and in this sense does not take into account the physical properties of the cortex. The harmonic and temporal power spectra, as well as the functional connectivity matrix, are shown in Fig 1 for the case with $h = 10^{-4}$m.

The presence of *fast*, long-range connectivity can impact the power spectrum and functional interactions of equilibrium fluctuations, as well as the stability of steady-states. To illustrate this, we add a single non-local edge between vertices 250 and 750 to the one-dimensional graph with $h = 10^{-4}$m. The Euclidean distance between these two vertices is $500 \cdot h = 5 \cdot 10^{-2}$m = 5cm. In the healthy brain, myelination allows activity propagation along white-matter fibers to take place at speeds $\sim 200$ times greater than local surface propagation [25]. To model myelination, we set the length of the non-local edge to be the Euclidean

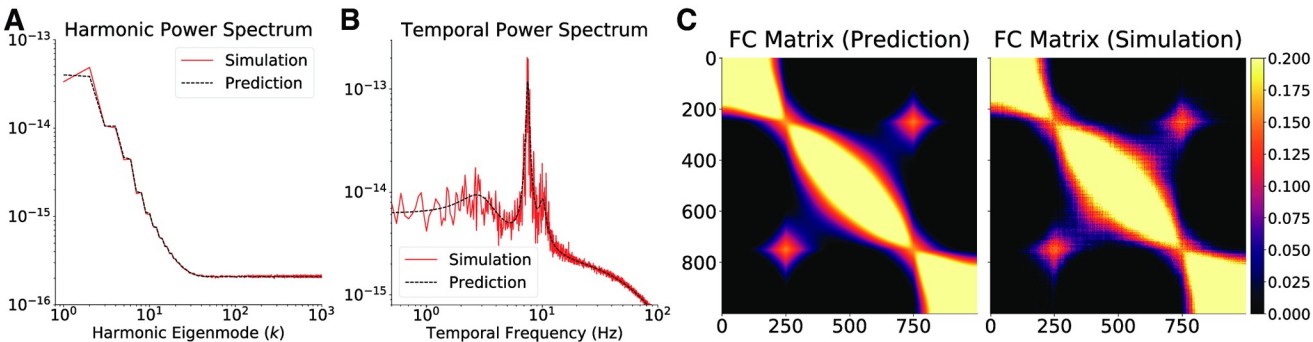

**Fig 4. Emergence of multiple temporal power peaks by long-range inhibition.** (A) The harmonic and (B) temporal power spectra of Excitatory activity equilibrium fluctuations in the one-dimensional graph, with the size of the Gaussian kernel controlling Inhibitory to Excitatory interactions $\sigma_{IE}$ increased by a factory of 20, and everything else unchanged with respect to Fig 3. Allowing Inhibitory activity to exert its influence over larger distances here leads to the emergence of multiple peaks in the temporal power spectrum of Excitatory activity. The dashed black lines correspond to the theoretical prediction and the red lines are obtained through numerical simulations. (C) The Excitatory activity functional connectivity obtained by analytic predictions and numerical simulations.

distance between the vertices, divided by a factor of 200 (similarly to the construction of the human connectome graph Laplacian, where the length of white-matter edges is set to be their 3D path-length distance along DTI fibers, divided by a factor of 200). Therefore, the effective length of the non-local edge is $2.5 \cdot 10^{-4}$m. Fig 2A and 2B shows the effects of the presence of the non-local edge on the harmonic and temporal power spectra of the equilibrium fluctuations. The most pronounced effect is damping of the oscillatory resonance in the temporal power spectrum, thus rendering the fluctuations more stable. Furthermore, the edge leads to a discernible alteration in the functional connectivity (Fig 2C).

Interestingly, when the model operates in the pathological i.e. non-stable regime ($h = 2 \cdot 10^{-4}$m), addition of a single non-local edge stabilizes the steady state, thus leading to healthy equilibrium fluctuations (Fig 3B). The non-local edge also creates a large increase in the functional connectivity between the vertices involved, and a change in the pattern in neighboring vertices (Fig 3C). As noted above, these effects of long-range connectivity are observed if the effective length of the non-local edge is small enough for non-local activity propagation to interact with local activity propagation. For these one-dimensional simulations, this happens if the speed factor is larger than $\sim 50$.

In Fig 4, we show an illustrative example of how kernel parameters can lead to significant alterations in observable model dynamics. Increasing the size of the Gaussian kernel controlling the Inhibitory to Excitatory interaction ($\sigma_{IE}$) by a factor of 20 leads to the emergence of multiple peaks in the temporal power spectrum of the model. Increasing the value further, for example by a factor of 30, renders the steady state unstable. All other parameters, presence of non-local edge, and distance-weighting were left unchanged with respect to Fig 3.

## Application to resting-state fMRI

To illustrate the applicability of graph neural fields, we study the stochastic Wilson-Cowan graph neural field on a single-subject multimodal connectome, and investigate whether the model can capture empirical observables of resting-state fMRI. The connectome is of mesoscopic resolution, comprising of approximately 18000 cortical surface vertices (MRI) and 15000 white matter tracts (DTI). See connectome construction for details on the construction of the weighted connectome graph Laplacian.

**Graph neural fields on the human connectome reproduce the harmonic power spectrum of resting-state fMRI.** First, we obtain the harmonic power spectrum of resting-state fMRI, according to its definition, as the temporal mean of the squared graph Fourier transform of the fMRI timecourses. Note that the estimation of the fMRI harmonic power spectrum does not use a single timepoint, but the entire available timecourse. To regularize the empirical spectrum, we compute its log-log binned median with 300 bins, following [26]; eigenmodes above $k = 15000$ contain artifacts due to reaching the limits of fMRI spatial resolution, and are thus removed. We optimize the model parameters with a basinhopping procedure [27], aiming to minimize the residual difference between empirical and theoretical harmonic power spectra. The parameter set producing the best-fit harmonic power spectrum is reported in S2 Table. In the fitting, we allow for a linear rescaling as a simple observation model connecting the theoretical and empirical spectra:

$$H_{\text{fMRI}}(k) = \beta H_E(k), \tag{15}$$

where $H_{\text{fMRI}}(k)$ is the harmonic power spectrum of the fMRI data, $H_E(k)$ is the analytically predicted harmonic power spectrum of Excitatory neural activity (Eq (14)), and $\beta$ is a linear rescaling parameter. To verify the accuracy of the analytic prediction, we carry out numerical simulations of the model Eqs (7 and 8) on the connectome, with a time-step value $\delta t = 10^{-4}$

# Harmonic Power Spectrum

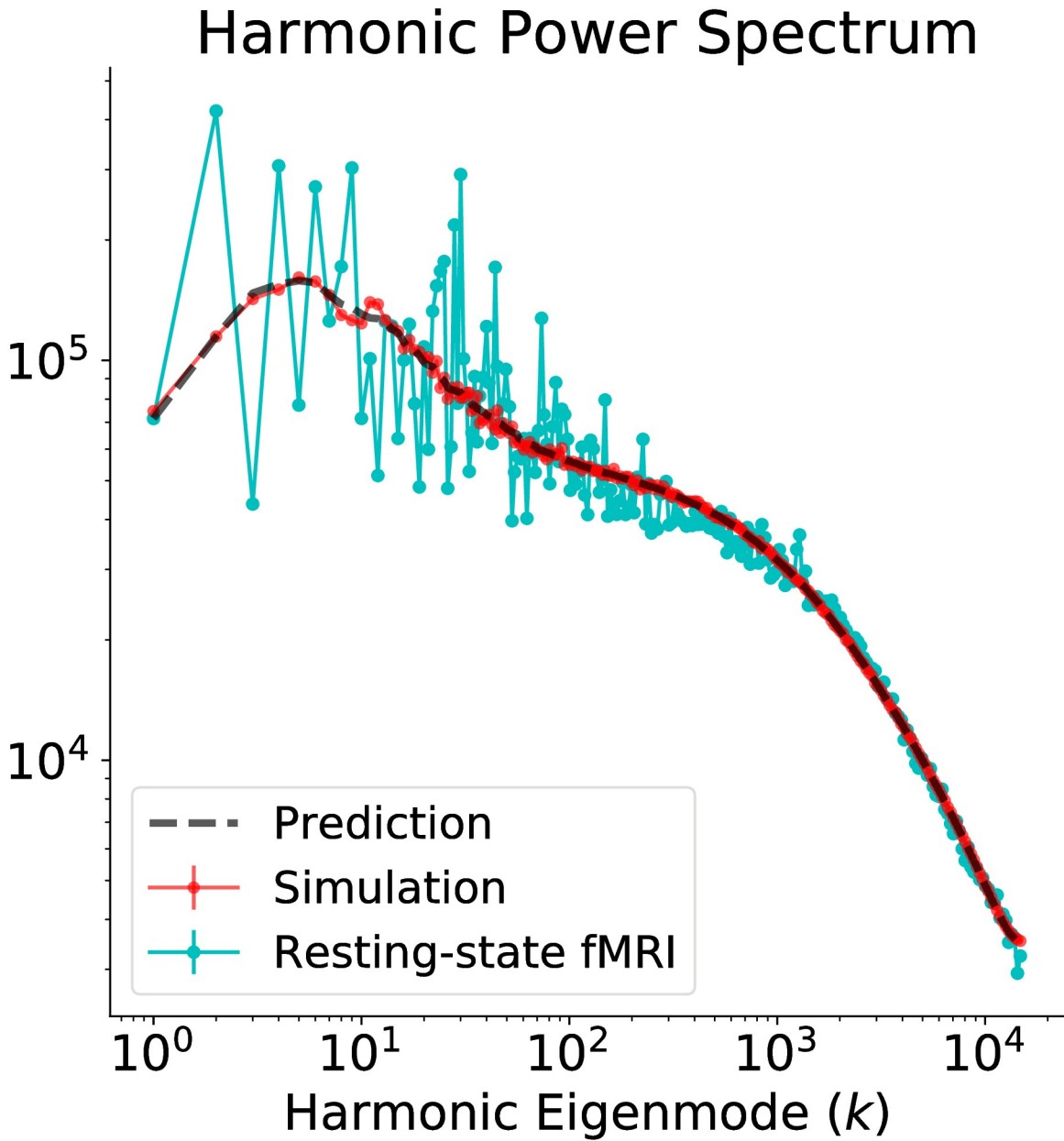

**Fig 5. Stochastic Wilson-Cowan graph neural field model captures the resting-state fMRI harmonic power spectrum.** The theoretical (dashed black line) and numerical (red line) predictions from the stochastic Wilson-Cowan graph neural field model, with the parameters of S2 Table, are in excellent agreement with the empirically observed fMRI harmonic spectrum (cyan line). The numerical spectrum was obtained by taking the median of three independent simulations.

seconds, and an observation time of $10^6$ time-steps, corresponding to 100 seconds of simulated activity. S5 Fig shows snapshots of the simulated model and of resting-state fMRI, at different times. Fig 5 shows the harmonic power spectra of fMRI data and of the stochastic Wilson-Cowan graph neural field model, with the parameter set of S2 Table. The model is clearly able to reproduce the fMRI harmonic power spectrum, showing excellent agreement between analytically predicted, numerically simulated, and empirically observed harmonic power spectra. Previous studies have shown that the harmonic power spectrum of resting-state fMRI can be

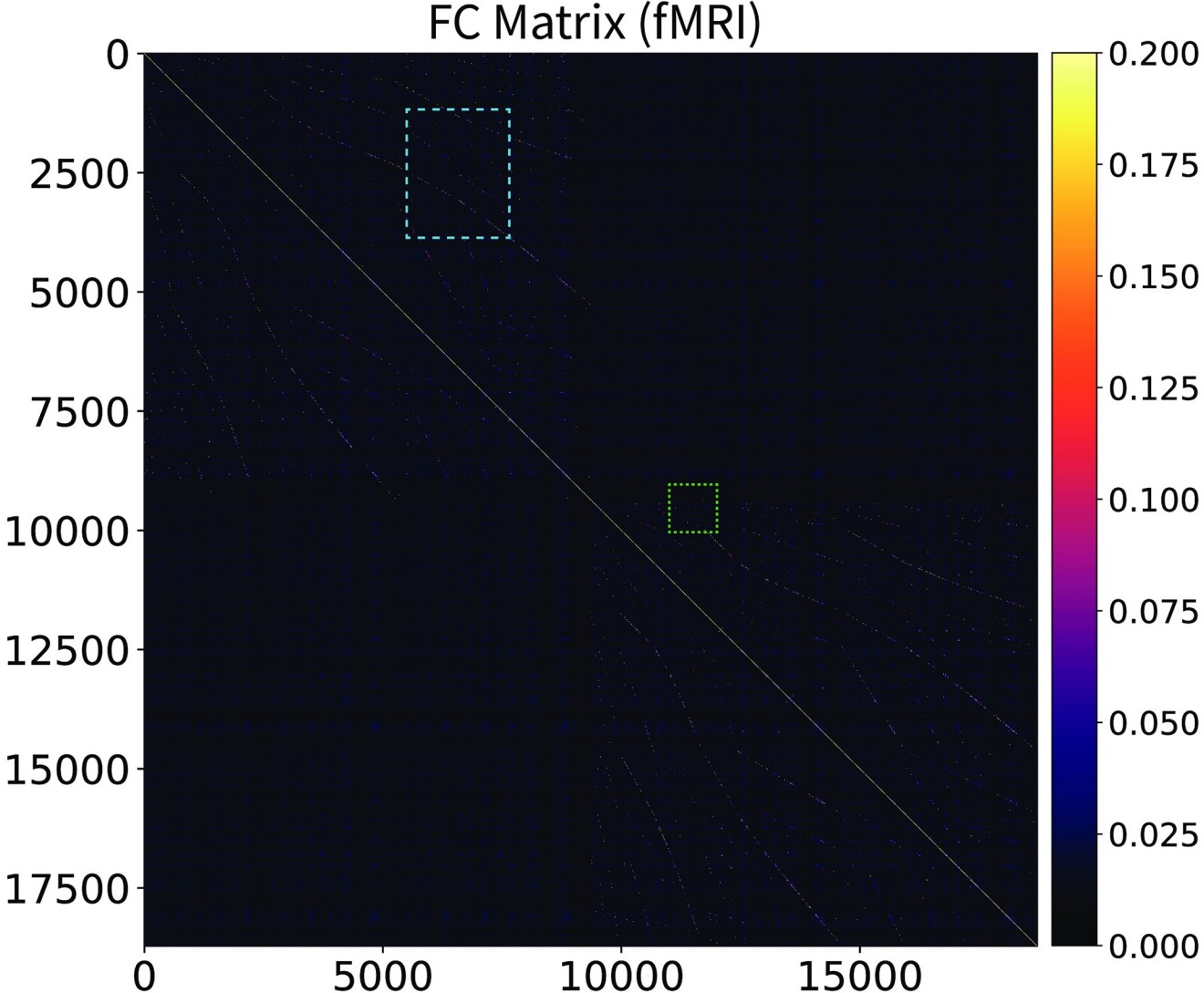

**Fig 6. Resting-state fMRI functional connectivity matrix.** Connectome-wide, vertex-wise, single-subject, resting-state fMRI functional connectivity matrix. Zoom in to appreciate the patterns present in the data, in particular the two large blocks (top-left and bottom-right) corresponding to the two hemispheres, and the many intra-hemispheric patterns. Compare with the functional connectivity predicted by the stochastic Wilson-Cowan graph neural field (Fig 7). The light-blue and light-green rectangles indicate the insets visualized in Figs 8 and 9.

used to differentiate between a placebo condition and the altered state of consciousness induced by a serotonergic hallucinogen, lysergic acid diethylamide (LSD) [26]. LSD is known to have profound effects on perception and cognition; furthermoe, together with other psyche-delic compounds, it is currently under investigation in the treatment of several psychiatric conditions [28–30]. Thus, the ability to reproduce the harmonic power spectrum of resting-state fMRI shows that graph neural fields are capable of capturing measures of neural dynam-ics relevant for brain function and clinical applications.

**Graph neural fields on the human connectome predict the vertex-wise functional con-nectivity of resting-state fMRI.** The CHAOSS method also provides an analytic prediction of the model functional connectivity (correlation) matrix (Eq (46)). In Figs 6 and 7 we com-pare the resting-state fMRI functional connectivity with the theoretical prediction from the

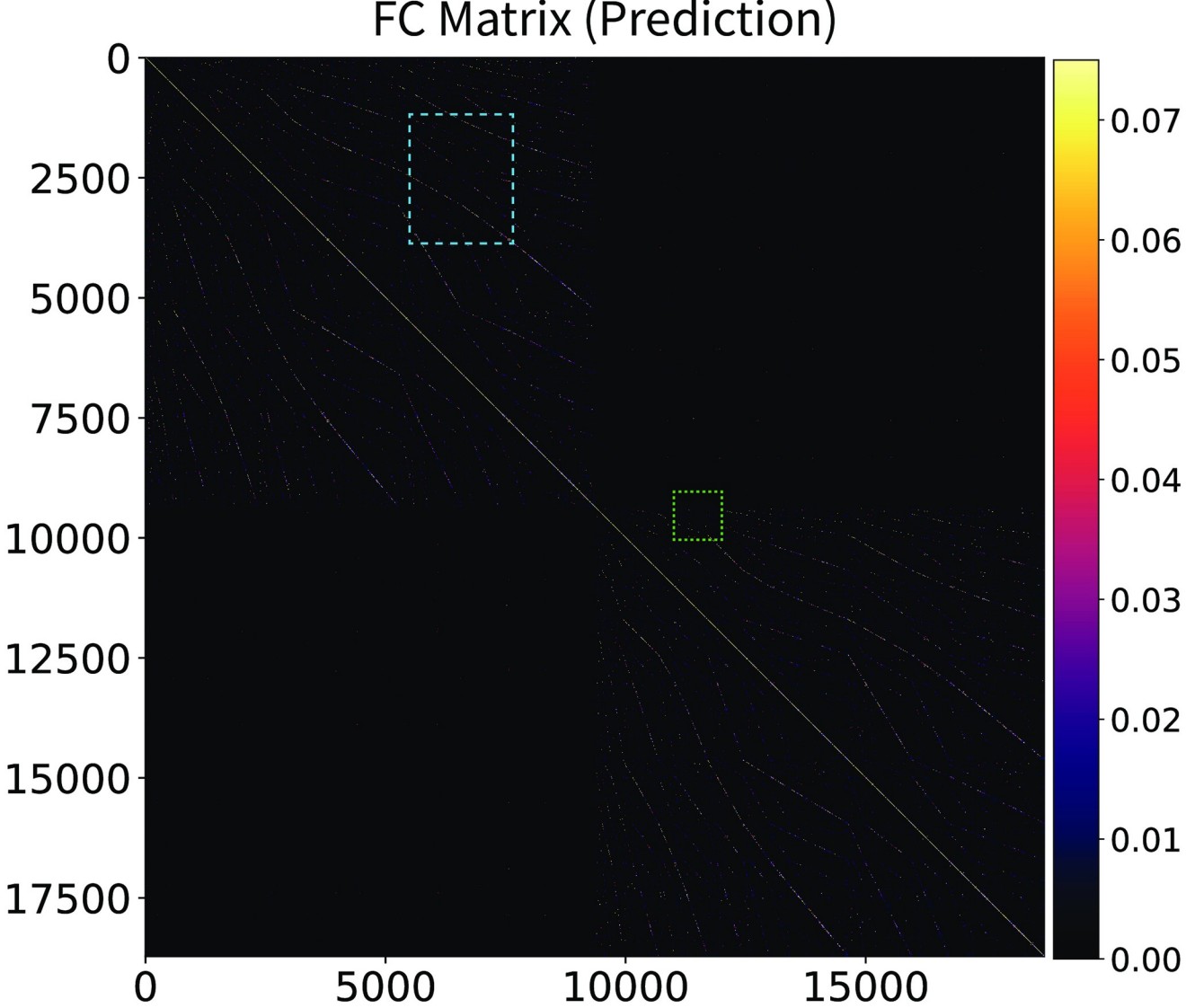

**Fig 7. Stochastic Wilson-Cowan graph neural field model predicts the experimental functional connectivity matrix.** The CHAOSS prediction for the connectome-wide, vertex-wise, single-subject functional connectivity matrix of the stochastic Wilson-Cowan graph neural field model with the parameters of S2 Table. Compare with Fig 6 to appreciate how the model predicts the patterns of functional connectivity observed in the fMRI data. The light-blue and light-green rectangles indicate the insets visualized in Figs 8 and 9. Note that we did not fit the fMRI functional connectivity of the model to the data, but only the harmonic power spectrum.

Wilson-Cowan graph neural field model with the parameters of S2 Table. The matrices are shown for the full connectome, at vertex-wise resolution, with no parcellation or smoothing. Vertex-wise fMRI functional connectivity on a connectome with 18000 vertices is naturally somewhat more noisy than the model analytic prediction, hence the choice of a slightly wider color-scale for the fMRI matrices, which emphasizes patterns in the data and deemphasizes background noise. Functional connectivity patterns in the empirical and theoretically predicted matrices are in clear agreement; two main blocks of connectivity can be distinguished, corresponding to the hemispheres, in the top-left and bottom-right of the matrices, as well as many corresponding intra-hemispheric features. In Figs 8 and 9, we show insets, at different scales, of the empirical and theoretical matrices. Because of the high number of vertices in the

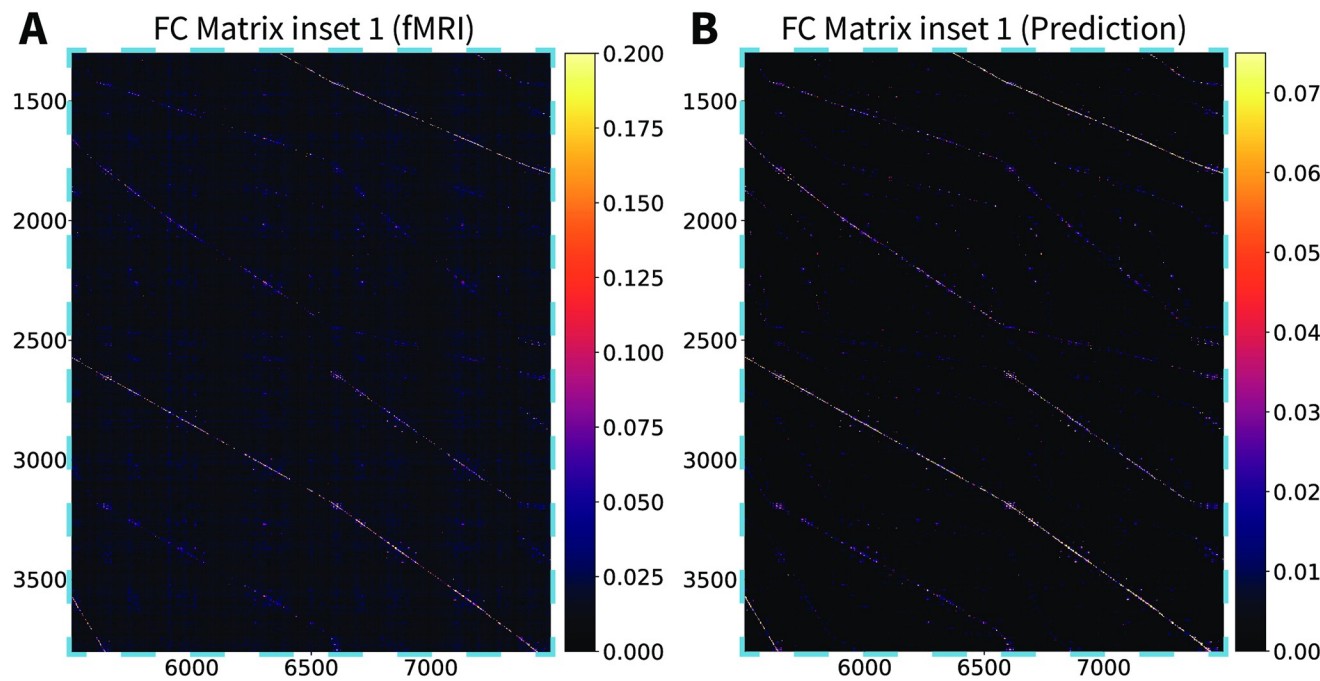

**Fig 8. Stochastic Wilson-Cowan graph neural field model predicts the experimental functional connectivity matrix (inset 1).** (A) An inset of the vertex-wise, resting-state fMRI functional connectivity matrix for a single subject. (B) The same inset for the Wilson-Cowan graph neural field model with the parameters of S2 Table.

connectome, we recommend looking at the connectivity matrices on-screen, at the highest possible resolution; high-fidelity PDF versions of these figures are provided in S6–S9 Figs. We remark that we did not fit the functional connectivity matrix of the model to the data, but only the harmonic power spectrum. Besides the success and applicability of the graph neural field

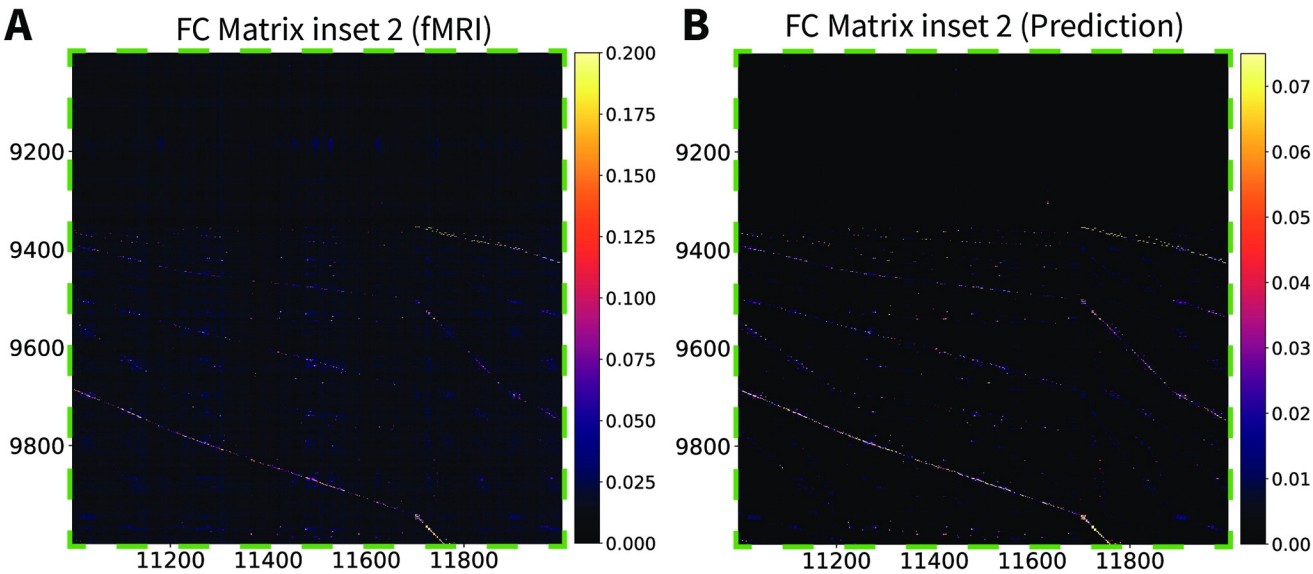

**Fig 9. Stochastic Wilson-Cowan graph neural field model predicts the experimental functional connectivity matrix (inset 2).** (A) An inset of the vertex-wise, resting-state fMRI functional connectivity matrix for a single subject. (B) The same inset for the Wilson-Cowan graph neural field model with the parameters of S2 Table.

approach, this result also demonstrates that the harmonic power spectrum is a robust measure of brain activity, capable of efficiently capturing features of neural dynamics with a high level of detail.

## Discussion

In this work, we have presented a general approach to whole-brain neural activity modelling on unparcellated, multimodal connectomes (*graph neural fields*), by combining tools from graph signal processing and neural field equations. We developed a technique to analytically compute observable quantities (CHAOSS). We showed that a Wilson-Cowan stochastic graph neural field model can reproduce the empirically observed harmonic spectrum of resting-state fMRI, and predict its functional connectivity matrix at vertex-wise resolution. Graph neural fields can address some limitations of existing modelling frameworks, and therefore represent a complementary approach resulting particularly suitable for mesoscopic-scale modelling and connectome-graph-based investigations. To discuss advantages and limitations of our approach, it is useful to contextualize it within the landscape of whole-brain models.

Existing whole-brain models can be broadly divided into two classes, according to whether they incorporate short-range local connectivity or not. *Region-based models* only take into account long-range connectivity between dozens or few hundreds of macroscopic ROIs, whereas *surface-based models* directly incorporate short-range local connectivity as well [31, 32]. It is furthermore possible to distinguish between *discrete* and *continuous* surface-based models. Discrete surface-based models are defined on a (highly-sampled) cortex and are therefore finite-dimensional. In several studies, region-based and discrete surface-based models are collectively referred to as *networks of neural masses* [21, 33, 34]. Continuous, surface-based models are better known as *neural field models*, are defined on the entire cortex, and are infinite-dimensional [32, 35, 36]. Mathematically, discrete surface-based models are finite-dimensional systems of ordinary differential equations, whereas neural field models are partial integro-differential equations.

Region-based models are constructed by parcellating the cortex into a number of regions-of-interest (ROIs), placing a local model in each ROI, and connecting them according to a given connectome (see [2, 21, 34] for reviews). The ROIs are usually obtained from structural or functional cortical atlases and the number of ROIs is in the order of a hundred or less. Region-based mass models are characterized by the type of local models and how they are connected i.e. if the connections are weighted or not, Excitatory or Inhibitory, and if transmission delays are incorporated. A wide variety of local models has been used in the literature, including neural mass models, self-sustained oscillators, chaotic deterministic systems, circuits of spiking neurons, normal-form bifurcation models, rate models, and density models [2, 35]. Region-based models have proven valuable in understanding various aspects of large-scale cortical dynamics and their roles in cognitive and perceptual processing, but they are limited in one important respect: they do not allow studying the spatiotemporal organization of cortical activity on scales smaller than several squared centimeters and their effects on large-scale pattern formation. This is due to the fact that the dynamics within ROIs are described by a single model without spatial extent. This prevents studying the mesoscopic mechanisms underlying a large class of cortical activity patterns that have been observed in experiments, including traveling and spiral waves, sink-source dynamics, as well as their role in shaping macroscopic dynamics [21]. This is a significant limitation, particularly because the role of mesoscopic spatiotemporal dynamics in cognitive and perceptual processing is increasingly being recognized and experimentally studied [37, 38]. Graph neural fields present the advantage of allowing explicit modelling of activity propagation dynamics with spatiotemporal convolutions and

graph differential equations on mesoscopic-resolution connectomes, thereby overcoming this limitation.

Whole-brain models that incorporate short-range connectivity are referred to as *surface-based* because they are generally defined either on high-resolution surface-based representations of the cortex [31, 39, 40] or on the entire cortex viewed as a continuous medium. We will refer to these types of models as *discrete* and *continuous* surface-based models, the latter of which are known as *neural field models* [24, 32, 36, 41]. Numerically simulating discrete surface-based models is much more computationally demanding than simulating region-based models, as the former typically have dimensions that are one to two orders higher than those of the latter. Numerically simulating neural field models is even more demanding and requires heavy numerical integration in combination with specific analytical techniques [42]. Moreover, simulating neural field models requires special preparation of cortical meshes to ensure accuracy and numerical stability. [39, 40, 43–45]. Graph neural fields have the advantage of being implementable directly on multimodal structural connectomes obtained from MRI and DTI, thereby minimizing anatomical approximations, and being limited in this sense only by the quality and resolution of the available structural data. The cortex in graph neural field need not be a flat or spherical manifold, but can reflect the specific anatomy of each subject, allowing in-depth analyses of the effects of individual anatomical differences on functional activity; such analyses can then be compared across subjects thanks to the common language provided by the connectome harmonics. Graph neural fields can take into account important physical properties of the cortex, such as folding, non-uniform thickness, hemispheric asymmetries, non-homogeneous structural connectivity, and white-matter projections, since all these anatomical features can be absorbed in the distance-weighted graph Laplacian. In particular, we note that the extension to connectomes including cortical thickness, hence allowing activity to propagate not only tangentially but also perpendicularly to the cortical surface, is of particular interest. Cortical layers can already be distinguished with ultra-high field fMRI, and are thought to subserve different functions [46]. The ability of graph neural fields to account for cortical thickness and layers in dynamical models of neural activity is therefore a promising property for future development [47].

For ease of exposition, here we have focused mainly on neural field models with purely spatial kernels. Although this might be sufficient for modelling wide-band activity such as BOLD fMRI, the large-scale organization of oscillatory activity as recorded with EEG and MEG sensitively depends on the propagation delays of action potentials through white-matter fiber tracts [48–50]. To model such delays, spatiotemporal kernels have been used in continues neural field models [32, 36, 51, 52]. It is possible to extend this approach to graph neural fields, by using spatiotemporal graph convolutions, rather than purely spatial convolutions. This yields graphs filters that are more general than those in Eq (26) in that they depend not only on the eigenvalues of the graph Laplacian, but also on temporal frequency (S1 Appendix). The proposed method to fit graph neural fields to experimentally observed harmonic power spectra or functional connectivity matrices (CHAOSS) is straightforward to generalize as well, since the only difference is the appearance of complex exponentials in the linearized model equations in the temporal Fourier domain. With this extension, graph neural fields allow for the formulation of any spatiotemporal neural field model on arbitrary metric graphs.

Graph neural fields come equipped with computationally efficient analytic and numerical tools. The CHAOSS method allows fast computation of quantities such as the harmonic-temporal spectra or connectivity matrices without resorting to numerical simulations, which are enormously more computationally expensive than the direct evaluation of analytic expressions. This implies that optimization of model parameters (for example to fit an observable quantity such as the harmonic spectrum, as we do here) can be carried out without the

computational burden of numerical simulations. Furthermore, linear or linearized graph neural field equations are diagonalized by the graph Fourier transform, allowing very efficient numerical simulations in the graph Fourier domain. For a graph neural field on a connectome with $n$ vertices, carrying out numerical simulations in the graph Fourier domain reduces the dimensionality by a factor of $n$, which is a vast improvement for high-resolution connectomes. Hence, graph neural field analysis (CHAOSS) and numerical simulations (linear or linearized models in the graph Fourier domain) can be carried out with a minimal amount of computational power.

Our approach presents several limitations. First, the CHAOSS method as presented here, and the dimensionality reduction of linear or linearized equations in the graph Fourier domain, require the model parameters to be space-independent. That is, model parameters are assumed to have the same value for all vertices in the connectome. This assumption was also used in previous studies of continuous neural fields [53], and in our case has the advantage of allowing mathematical analyses and simulations that, as mentioned above, are scalable to higher-resolution connectomes with little computational cost. However, there are more biophysically realistic models that require space-dependent parameters. For example, some recent neural mass network models incorporate neuronal receptors and their densities, which are known to vary across the cortex [54–56]. The CHAOSS method can in principle be extended to account for space-dependent model parameters, and numerical simulations of graph neural fields can also be carried out with space-dependent parameters, but both would become significantly more computationally demanding than their counterparts with space-independent parameters. A possible approach to preserve computational efficiency, while characterizing regional differences, could be to absorb all the relevant space-dependent information into the graph Laplacian, maintaining space-indepedent model parameters. Similarly to the idea of differentially weighing white matter edges to account for myelination, one might weigh differentially graph edges within specific ROIs or specific subsets of vertices. Second, it is important to point out that our approach is subject to the limitations of tractography in regards to false positive and true negatives; and that the connectome used here does not include subcortical structures, nor projections between the cortex and subcortical structures. Future studies could attempt to employ connectomes including subcortical structures and connections. Third, the formulation of convolutions on graphs presented here is restricted to spatially symmetric kernels (but see the caption of S1 Fig for some considerations on indirect ways to obtain asymmetric kernels). Finally, another important limitation is the use of an undirected and time-independent connectome graph. For maximal generality and biophysical realism, one might want to study a directed, or even time-dependent (plastic) structural connectome. Such extensions would be very challenging, if at all feasible.

Immediate applications of graph neural fields can be found in the comparison of harmonic spectra, functional connectivity, and coherence matrix with single-subject empirical data obtained from different neuroimaging modalities such as fMRI and MEG, as well as different conditions, for example health, pathology, and neuropharmacologically-induced altered states of consciousness [26]. Investigating the effects of a reduced myelination speed factor, or pruned white-matter fibers, could be an interesting approach to modelling the effects of pathological or age-related structural alterations of white matter on the dynamics of functional activity. Other possible developments include the implementation of more biophysically realistic models, potentially including space-dependent parameters, and the use of a cortical connectome that includes cortical thickness, accounting for activity propagation across layers perpendicularly to the surface. Aside from whole-brain resting-state modelling, graph neural fields may also be used for modelling specific ROIs and stimulus-evoked brain activity. In particular, because of the known retinotopic mapping between visual stimuli and neural activity, the

visual cortex presents itself as a very interesting ROI for such developments [57]. Moving beyond neuronal populations and even the human brain, the mathematical framework of graph neural fields may also be used to implement single-neuron models directly on the full connectome graphs of simple organisms, such as *C. Elegans*, whose neuronal pathways have been experimentally mapped at the single-neuron level [22].

## Conclusion

In summary, in this study we described a class of whole-brain neural activity models which we refer to as *graph neural fields*, and showed that they can be used to capture dynamics of brain activity obtained from neuroimaging methods efficiently and with a high level of detail. The formulation of graph neural fields relies on existing concepts from the field of graph signal processing, namely the distance-weighted graph Laplacian operator and graph filtering, in combination with modelling concepts such as neural field equations. This framework allows inclusion of realistic anatomical features, analytic predictions of harmonic-temporal power spectra, correlation, and coherence matrices (*Connectome-Harmonic Analysis Of Spatiotemporal Spectra*, CHAOSS), and efficient numerical simulations. We illustrated the practical use of the framework by reproducing the harmonic spectrum and predicting the functional connectivity of resting-state fMRI with a stochastic Wilson-Cowan graph neural field model. Future work could build on the methods and results presented here, both from theoretical and applied standpoints.

## Methods

### Laplacian operators on graphs

In this section we provide a derivation of the distance-weighted graph Laplacian, or simply graph Laplacian, in terms of graph differential operators. The distance-weighted graph Laplacian is distinguished from the combinatorial graph Laplacian often used in analysis studies [11], as it allows geometrical properties of the cortex to be taken into account, which is necessary to implement physically realistic graph neural field models.

**The combinatorial Laplacian.** Consider an undirected graph with $n$ vertices. The binary adjacency matrix $\tilde{A}$ is defined as:

$$\tilde{A}_{ij} = \begin{cases} 1 & \text{if } i \sim j, \\ 0 & \text{otherwise.} \end{cases} \tag{16}$$

where $i \sim j$ means that vertices $i$ and $j$ are connected by an edge. The graph's degree matrix $\tilde{D}$ is a diagonal matrix whose diagonal entries are given by:

$$\tilde{D}_{ii} = \sum_{j=1}^{n} \tilde{A}_{ij}. \tag{17}$$

It hence counts the number of edges for each vertex $i$. The binary or *combinatorial graph Laplacian*, denoted by $\tilde{\Delta}$, is defined as:

$$\tilde{\Delta} = \tilde{A} - \tilde{D}. \tag{18}$$

The combinatorial graph Laplacian and its normalized version do not carry information about the distances between cortical vertices and therefore are invariant under topological but non-isometric deformations of graph. Neural activity modeled in terms of the combinatorial graph Laplacian therefore is a topological graph invariant, whereas real neural activity does

depend on the metric properties of the graph. The combinatorial graph Laplacian, however, can be adjusted so as to take into account the metric properties of the graph, yielding the *distance-weighted graph Laplacian*. Below, we provide a derivation of the weighted graph Laplacian in terms of the graph directional derivaties of a graph function.

**The distance-weighted graph Laplacian.** Let $f$ be a a function defined on the vertices of a graph, and let $M$ be the graph's distance matrix. Thus, the $(i, j)$ entry $M_{ij}$ of $M$ equals the distance between vertices $i$ and $j$ in a particular metric. We note that for this derivation, it is irrelevant how $M$ is obtained. In the context of connectomes, the elements of $M$ can be defined in terms of suitably scaled Euclidean distances, geodesic distances over the cortical manifold, or as the lengths of white matter fibers connecting vertices. Different distance metrics can also be combined for the construction of connectome graphs containing multiple types of edges, as we do here (see Data preprocessing and connectome graph construction), and as has been done in some previous studies [58]. The first-order graph directional derivative $\partial_j f_i$ of $f$ at vertex $i$ in the direction of vertex $j$ is:

$$\partial_j f_i = \frac{\tilde{A}_{ij}}{M_{ij}} (f_j - f_i). \tag{19}$$

Note that according to this definition, $\partial_j f_i = 0$ if vertex $j$ is not connected to vertex $i$, and that $\partial_i f_i = 0$. Also note that $\partial_j$ is a linear operator on the vector space of graph signals. Furthermore, since $\tilde{A}_{ij}^2 = \tilde{A}_{ij}$, the second-order graph directional derivative $\partial_j^2 f_i$ of $f$ at vertex $i$ in the direction of vertex $j$ is defined as:

$$\partial_j(\partial_j f_i) = \partial_j^2 f_i = \frac{\tilde{A}_{ij}}{M_{ij}^2} (f_i - f_j). \tag{20}$$

Following the definition of the Laplacian operator in Euclidean space as the sum of second-order partial derivatives, the *distance-weighted graph Laplacian*, or simply *graph Laplacian* $\Delta$ is defined as:

$$\Delta f_i = -\sum_{j=1}^{n} \partial_j^2 f_i. \tag{21}$$

To see the relation with the combinatorical graph Laplacian, we note that $\Delta$ can be written in matrix form as:

$$\Delta = A - D, \tag{22}$$

where $A$ and $D$ are the distance-weighted adjacency matrix and distance-weighted degree matrix, respectively, which are defined as $A_{ij} = \tilde{A}_{ij}/M_{ij}^2$ and $D_{ii} = \sum_{j=1}^{n} A_{ij}$, respectively. Thus, the weighted graph Laplacian can be obtained by using the weighted versions of the adjacency and degree matrices in the definition of the combinatorial graph Laplacian.

**The graph Fourier transform.** Diagonalization of the graph Laplacian gives:

$$\Delta = U\Lambda U^T, \tag{23}$$

where $U$ is an orthogonal matrix containing the eigenvectors of $\Delta$, and $\Lambda$ is a diagonal matrix containing the corresponding eigenvalues $\lambda_1 \geq \lambda_2 \geq, \cdots, \geq \lambda_n \geq 0$. The graph Fourier transform of a function $u(t)$ on the graph is defined by:

$$\hat{u}(t) = U^T u(t), \tag{24}$$

where the transformation $U^T$ expresses $u(t)$ in the eigenbasis of $\Delta$. The vertex-domain signal $u(t)$ can be recovered again by applying the inverse graph Fourier transform $U^{-1} = U$ to $\hat{u}(t)$. For clarity, note that the graph Fourier transform is not related to the temporal Fourier transform and that $u(t)$ does not have to depend on time to apply it. For grid graphs (i.e. graphs whose drawing, embedded in some Euclidean space, forms a regular tiling), the graph Fourier transform is equivalent, in the continuum limit, to the spatial Fourier transform in Euclidean space. However, the graph Fourier transform can also be applied to more complex graphs, possibly with non-local edges, such as the human connectome.

## Convolution kernels on graphs

In order to define neural field equations on graphs, we need a graph-theoretical analog of the continuous spatial convolution:

$$(K \otimes u)(x, t) = \int_{-\infty}^{\infty} K(x - x')u(x', t)dx'. \qquad (25)$$

To obtain this, we use the convolution theorem to represent the convolution in the spatial Fourier domain as $\hat{K}(k)\hat{u}(k, t)$, where $k$ is the spatial wavenumber. When the kernel is real-valued and spatially symmetric, its Fourier transform is real-valued and even in $k$, so that $\hat{K}(k)$ can be viewed as a function of $-k^2$. In continuous space, $-k^2$ is the eigenvalue of the spatial Fourier basis function $e^{ikx}$ under the Laplace operator. On graphs, the distance-weighted graph Laplacian $\Delta$ implements a generalized version of the Laplace operator, and the graph Fourier basis is defined by its eigenvectors $U$. Hence, the graph filter $\hat{K}_g$ corresponding to the convolution $(K \otimes u)(x, t)$ can be defined by substituting $\lambda_k$ for the values $-k^2$ in $\hat{K}(-k^2)$:

$$\hat{K}_g = \text{Diag}(\hat{K}(\lambda_1), \cdots, \hat{K}(\lambda_n)). \qquad (26)$$

In the graph Fourier domain, the filtered (convolved) signal is hence per definition given by:

$$\hat{u}^{\text{filt}}(t) = \hat{K}_g \hat{u}(t). \qquad (27)$$

Applying the inverse graph Fourier transform $U$, we obtain the filtered signal in the graph domain:

$$u^{\text{filt}}(t) = U\hat{K}_g\hat{u}(t) = U\hat{K}_g U^T u(t) = K_g u(t), \qquad (28)$$

where we have defined $K_g = U\hat{K}_g U^T$, the graph domain representation of the filter. Eqs (27 and 28) can be interpreted as an analogy for the convolution theorems on graphs: the matrix-multiplication implementing a convolution in the graph domain becomes a point-wise product in the graph Fourier domain, since $\hat{K}_g$ is a diagonal matrix. This analogy can be employed to define spatiotemporal convolutions (S1 Appendix), and reaction-diffusion models (S2 Appendix), on arbitrary metric graphs. For example, the damped-wave and telegrapher's equations (S3 Appendix), of interest in the context of modelling the propagation of neural signals, can be implemented on the human connectome (S2–S4 Figs).

**Examples of graph kernels.** Table 2 lists several commonly used continuous spatial kernels and their equivalent filter on graphs. On grid graphs, the filters simply act as discretized versions of their continuous counterparts. However, this approach generalizes to arbitrary metric graphs, potentially with non-local edges, such as the human connectome, and is therefore more broader in scope than grid-based discretizations of continuous convolution kernels.

## Graph neural fields

Continuous neural field models describe the dynamics of cortical activity $u(x, t)$ at time $t$ and cortical location $x \in \Omega$. Here, $\Omega \in \mathbb{R}^3$ denotes the cortical manifold embedded in three-dimensional Euclidean space. Depending on the physical interpretation of the state variable $u(x, t)$, neural fields come in two types, which we will refer to in the rest of the text as *Type 1* and *Type 2*. This short description is by no means meant to be exhaustive, and only contains the required background to define graph neural fields; comprehensive treatments of continuous neural fields are provided in [36, 53].

In Type 1 neural fields [53], the state variable $u(x, t)$ describes the average membrane potential at location $x$ and time $t$. The general form of a neural field model of Type 1 is:

$$D_t u(x, t) = \int_\Omega K(d(x, x')) S[u(x', t)] dx' + \sigma \xi(x, t), \qquad (29)$$

where $\sigma \xi(x, t)$ is the noise term, $d(x, x')$ is the geodesic distance between cortical locations $x$ and $x'$, $K$ is the spatial kernel of the neural field that describes how the firing-rate $S[u(x', t)]$ at location $x'$ affects the voltage at location $x$, and S is the firing-rate function that converts voltages to firing-rates. $D_t$ is a placeholder for the linear temporal differential operator that models synaptic dynamics, and can take different forms depending on the model under investigation. In modelling resting-state cortical activity, $\xi(x, t)$ is usually taken to be a stationary stochastic process. For simplicity, we will assume the stochastic term $\xi(x, t)$ to be spatiotemporally white noise (but in principle, colored noise could be used as well). The distance function $d(x, x')$ between cortical locations $x$ and $x'$, as well as the integration over the cortical manifold $\Omega$, assume that $\Omega$ is equipped with a Riemannian metric. A natural choice is the Euclidean metric induced by the embedding of the cortical manifold in three-dimensional Euclidean space.

In Type 2 neural field models [59, 60], the state variable $u(x, t)$ denotes the fraction of active cells in a local cell population at location $x$ and time $t$, and hence takes values in the interval $[0, 1]$. Type 2 neural field models have the form:

$$D_t u(x, t) = S\left[\int_\Omega K(d(x, x')) u(x', t) dx'\right] + \sigma \xi(x, t), \qquad (30)$$

where S denotes the activation function that maps fractions to fractions and hence takes values in the interval $[0, 1]$ and thus has a different interpretation from the firing-rate function in Type 1 neural field models. Mathematically, the only difference between Type 1 and Type 2 neural field models is the placement of the non-linear function S. In practice, most neural field models are defined by two or more neural field equations, where each equation describes the dynamics of a different neuronal population, and its interaction with the other cell types. For example, the state variable of the Wilson-Cowan neural field model (Eqs (1 and 2)) is two-dimensional and its components correspond to Excitatory and Inhibitory neuronal populations.

In theoretical studies on neural field models, the cortex is usually assumed to be flat:, i.e. $\Omega = \mathbb{R}^2$ (cortical sheet) or $\Omega = \mathbb{R}^1$ (cortical line) or a closed subset thereof (but see [61] for a detailed theoretical study of a neural field model on the sphere). The major simplification that occurs in this case is that the cortical metric reduces to the Euclidean metric:

$$d(x, x') = \|x - x'\|, \qquad (31)$$

and, as a consequence, the integrals in Eqs (29) and (30) reduce to spatial convolutions, so that Fourier methods can be used in the analysis. For spatially symmetric kernels, i.e. $K(-x) = K(x)$

for all $x \in \mathbb{R}$, convolutions integrals can be translated to graphs using the methods of the previous section Convolution kernels on graphs.

Thus, a *graph neural field of Type 1* is a model of the form:

$$D_t u(t) = K_g S[u(t)] + \sigma \xi(t), \tag{32}$$

and a *graph neural field of Type 2* is a model of the form:

$$D_t u(t) = S[K_g u(t)] + \sigma \xi(t). \tag{33}$$

When more than one type of neuronal population is included, as for the Wilson-Cowan model, or when the temporal differential operator $D_t$ is of order higher than one, the continuous neural fields reduce to *systems* of ordinary differential equations on graphs.

The continuous neural fields in Eqs (29) and (30) are described by partial integro-differential equations in which the integration in done over space. Continuous neural fields can also be described by spatiotemporal integral equations by viewing the temporal differential operator $D_t$ as a temporal integral, which leads to a more general class of models. By defining spatiotemporal convolutions on graphs (S1 Appendix), this larger class of neural fields can be formulated on graphs as systems of temporal integral equations. To make this explicit, we use the definition of the spatiotemporal graph filtering operator $K_g \otimes$ to write out the $i^{th}$ component of $u$, for a neural field of Type 1:

$$u_i(t) = \int_{-\infty}^{\infty} \left( \sum_{j=1}^{n} K_g^{ij}(s) S[u_j(t-s)] \right) ds + \sigma \int_{-\infty}^{t} \xi_i(t') dt'. \tag{34}$$

Thus, the spatiotemporal integrals in continuous neural fields are replaced by temporal integrals in graph neural fields, and the spatial structure of the continuous kernel is incorporated into the graph filter $K_g^{ij}$. The same applies to neural fields of Type 2. Furthermore, for separable kernels, and for special choices of the temporal component of the kernel, the spatiotemporal integral equation can be reduced to a partial integro-differential equation [32, 36]. For graph neural fields there exists an equivalent subset of models that can be represented by a system of ordinary integro-differential equations.

Eqs (32 and 33) define graph neural fields for the case of purely spatial kernels $K(x, t) = K(x)$. In case of a purely temporal kernel $K(x, t) = g_\Theta(t)$, we obtain the following systems of ordinary differential equations, for a graph neural field of Type 1:

$$D_t u(t) = (g_\Theta \otimes S[u])(t) + \sigma \xi(t), \tag{35}$$

and for a graph neural field of Type 2:

$$D_t u(t) = S[(g_\Theta \otimes u)(t)] + \sigma \xi(t). \tag{36}$$

In case of a separable kernel $K(x, t) = w(x) g_\Theta(t)$ we obtain the following systems of ordinary differential equations, for a graph neural field of Type 1:

$$D_t u(t) = (g_\Theta \otimes K_g S[u])(t) + \sigma \xi(t), \tag{37}$$

and for a graph neural field of Type 2:

$$D_t u(t) = S[(g_\Theta \otimes K_g u)(t)] + \sigma \xi(t). \tag{38}$$

### Relating graph neural fields to experimental observables

**Connectome-Harmonic Analysis Of Spatiotemporal Spectra (CHAOSS).** To characterize the spatiotemporal dynamics of resting-state brain activity, we derive analytic predictions for harmonic and temporal power spectra, functional connectivity, and coherence matrices of graph neural fields. For simplicity, we carry out the derivation for the case of space-independent model parameters. It is possible to extend the method to the case with space-dependent parameters, but all computations would be significantly more burdensome. For graph neural fields with space-independent parameters, the linear or linearized model equations for each graph Laplacian eigenmode can be described as the following $p$-dimensional system, where $p$ is the number of neuronal population types:

$$D_t \hat{u}_k(t) = J_k \hat{u}_k(t) + \sqrt{B} \hat{\xi}_k(t). \tag{39}$$

Taking the temporal Fourier transform we obtain:

$$\hat{u}_k(\omega) = [D(\omega) - J_k]^{-1} \sqrt{B} \hat{\xi}_k(\omega), \tag{40}$$

where $D(\omega)$ denotes the temporal Fourier transform of $D_t$. Abbreviating the graph filter $\hat{K}_g = [D(\omega) - J_k]^{-1}$, the cross-spectral matrix $S_k(\omega)$ of the $k^{th}$ eigenmode is given by:

$$S_k(\omega) = \mathbb{E}[\hat{u}_k(\omega) \hat{u}_k(\omega)^\dagger] = \hat{K}_g \sqrt{B} \mathbb{E}[\hat{\xi}_k(\omega) \hat{\xi}_k(\omega)^\dagger] \sqrt{B} \hat{K}_g^\dagger = \hat{K}_g B \hat{K}_g^\dagger, \tag{41}$$

where † denotes the conjugate transpose and $\mathbb{E}$ denotes the expected value. Colored noise can be modeled by letting $B$ depend on $\omega$, although this is usually not done in neural field modelling studies. Another possible generalization is to let $B$ depend on the harmonic eigenmode.

Eq (41) gives a closed-form expression for the cross-spectral matrix of the $k^{th}$ eigenmode. Hence, its $s^{th}$ diagonal entry $[S_k(\omega)]_s$, with $s = 1, \ldots, p$, represents the power of the $s^{th}$ neuronal population, in the $k^{th}$ eigenmode, at temporal frequency $\omega$. The temporal power spectrum $T_s(\omega)$ of the $s^{th}$ neuronal population is obtained by summing over harmonic eigenmodes:

$$T_s(\omega) = 2 \sum_{k=1}^{n} [S_k(\omega)]_s, \tag{42}$$

where the factor of 2 arises because on graphs, $k$ ranges only over positive integers between 1 and $n$. Similarly, the harmonic power spectrum of the $s^{th}$ neuronal population $H_s(k)$ is obtained by integrating over the temporal frequency $\omega$:

$$H_s(k) = \frac{1}{2\pi} \int_{-\infty}^{+\infty} [S_k(\omega)]_s d\omega. \tag{43}$$

When combined with a suitable observation model, these predictions can be compared with or fitted to experimental data from different neuroimaging modalities.

**Functional connectivity.** Furthermore, it is possible to compute the correlation matrix of brain activity for each neuronal population. To construct the covariance matrix of a neuronal population activity $\Sigma_s$ across all graph vertices, we first construct the covariance matrix $\hat{\Sigma}_s$ in the graph Fourier domain. The covariance matrix of the $s^{th}$ neuronal population in the graph Fourier domain $\hat{\Sigma}_s$ is given by:

$$\hat{\Sigma}_s = \text{Diag}(H_s(k)). \tag{44}$$

The covariance matrix across all vertices is obtained by transforming back to the graph domain:

$$\Sigma_s = U\hat{\Sigma}_s U^T. \tag{45}$$

The functional connectivity (correlation) matrix $F_s$, which is often used in fMRI resting-state studies, is obtained by normalizing the covariance matrix, so that its entries are in the range $[-1, 1]$:

$$F_s = (\Sigma_s^+)^{-1/2}\Sigma_s(\Sigma_s^+)^{-1/2}, \tag{46}$$

where $\Sigma_s^+$ denotes $\Sigma_s$ with all off-diagonal entries set to zero. Seed-based connectivity of the $j^{th}$ vertex is measured by the $j^{th}$ row (or column) of $F_s$. Eq (46) provides an analytic prediction for the vertex-wise functional connectivity of graph neural fields.

**Coherence matrix.** From the linearized model equations one can also derive the coherence matrix, which measures the strength and latency of interactions between pairs of vertices as a function of frequency $\omega$, and is often used in EEG and MEG studies [62]. If the noise is assumed to be white, non-linear connectivity measures such as the phase-locking value and amplitude correlations can be analytically computed from the coherence matrix [63]. For simplicity, we derive the coherence matrix for the case of a single neuronal population and space-independent parameters.

The derivation of the coherence matrix is similar to that of the functional connectivity, and starts by expressing the linearized model equations in the vertex domain:

$$D_t u(t) = Ju(t) + \sqrt{B}\hat{\xi}(t). \tag{47}$$

Transforming Eq (47) to the temporal Fourier domain and taking expectations yields the cross-spectral matrix $S_v(\omega)$ in the vertex domain:

$$S_v(\omega) = \mathbb{E}[u(\omega)u(\omega)^\dagger] = K_g B K_g^\dagger, \tag{48}$$

where $K_g = [D(\omega) - J]^{-1}$. The coherence matrix $C(\omega)$ is obtained by normalization of the cross-spectral matrix in the vertex domain:

$$C(\omega) = (S_v^+(\omega))^{-1/2}S_v(\omega)(S_v^+(\omega))^{-1/2}, \tag{49}$$

where $S_v^+(\omega))$ denotes $S_v(\omega)$ with its off-diagonal entries to zero. The $(i, j)$ entry of $C(\omega)$ is the coherence between the cortical activity at vertices $i$ and $j$.

## Data preprocessing and connectome graph construction

We use structural MRI and DTI data obtained from the Human Connectome Project (https://db.humanconnectome.org/) to construct the individual subject anatomical connectome graph. In short, MRI data is employed to obtain local graph edges based on the surface mesh; DTI data is employed to add long-range white-matter connections to the graph. The main difference with previous studies analyzing brain activity in terms of the anatomical connectome graph Laplacian [11] is that instead of constructing the combinatorial (binary) graph Laplacian, here we construct a distance-weighted graph Laplacian (Eqs (19–22)). This allows us to take into account physical distance properties of the cortex that are relevant for graph neural fields, and that are otherwise lost. Specifically, for an local surface edge between vertices $i$ and $j$, the element $M_{ij}$ of the distance matrix $M$ is defined as their 3D Euclidean distance; for a non-local white-matter edge, $M_{ij}$ is defined as the distance along the respective DTI fiber path, divided by a factor of 200. This value is chosen to reflect the myelination of white matter fibers,

which is known to allow neural activity to propagate at speeds $\sim 200$ times greater in white matter fibers, in comparison with local surface propagation [25]. Resting-state BOLD fMRI timecourses from the Human Connectome Project were minimally preprocessed (coregistration, motion correction), resampled on the respective subject connectome graph, and demeaned.

## Supporting information

**S1 Table. Parameter set for 1D analysis and simulations.** This parameter set was obtained by a qualitative comparison of the Wilson-Cowan model's harmonic and temporal spectra with empirical data, and used to illustrate how graph properties affect neural field dynamics in one dimension.
(PDF)

**S2 Table. Parameter set for connectome-wide analysis and simulations.** This parameter set was obtained by quantiatively fitting the Wilson-Cowan model's harmonic power spectrum to that of resting-state fMRI data, and used for all connectome-wide analysis and numerical simulations.
(PDF)

**S1 Fig. Spatial convolution examples on 1-dimensional graphs.** To illustrate spatial convolution on graphs, we apply different spatial convolution filters from Table 2 to an impulse function centered on the middle vertex of a one-dimensional grid-graph with spacing $h = 1$ units. The resulting functions, normalized to have unit amplitude, show the shapes of the graph kernels. Note that the rectangular kernel convolution operator in Panel (E) exhibits the *Gibbs phenomenon* [64], which is a known feature of finite Fourier representations of functions with jump discontinuities. Solutions to this problem have been offered [65], but they are beyond the scope of the current work. Thus, we suggest avoiding spatial kernels with jump discontinuities in the context of graph neural fields. Open boundary conditions can be implemented by extending the graph beyond the image size, and periodic boundaries by adding edges connecting vertices on opposite sides of the graph. We also note that, if desired, spectral kernels can be obtained using polynomial approximation schemes, which obviates the need to diagonalize the graph Laplacian matrix [66]. For large datasets (for example natural images databases), it might be computationally advantageous to apply convolutions with symmetric kernels through graph filters, rather than with standard discrete convolution methods. Blurring/ smoothing a 2-dimensional image with a spatial Gaussian kernel is equivalent to applying the graph Gaussian kernel to the image-function defined on a 2-dimensional square-grid graph. Spatial convolutions on graphs become linear matrix-vector products, which are highly optimized and easily parallelizable operations; the bulk of the computational cost for graph convolutions consists in the initial computation of the filter itself, which has to be performed only once per kernel. The approach described here is limited to symmetric kernels. In some special cases, asymmetric kernels may be practically obtained by introducing suitable asymmetries in the graph edges. For example, consider a grid graph in two dimensions, with additional edges connecting bottom-left and top-right vertices of each square in the grid. Because of the broken lattice symmetry, a Gaussian kernel on this non-grid graph will behave like a spatially elliptic Gaussian, angled at 45 degrees, analogously to modelling a spatially asymmetric diffusion process on the graph.
(TIF)

**S2 Fig. The damped-wave equation on the human connectome gives rise to propagation with characteristic speed and wavelength.** Shown are snapshots of simulated cortical activity

that is governed by the damped wave equation with time-step $\delta t = 1$ and parameters $a = 3 \cdot 10^5$, $b = 5 \cdot 10^3$.
(TIF)

**S3 Fig. Varying the parameters of the damped-wave equation alters the dynamics of propagation on the human connectome.** Shown are snapshots of simulated cortical activity that is governed by the damped wave equation with time-step $\delta t = 1$ and parameters $a = 1.5 \cdot 10^5$, $b = 2.5 \cdot 10^3$.
(TIF)

**S4 Fig. Dynamics of the damped-wave equation on the human connectome include non-local propagation along white-matter fibers.** Shown are snapshots of simulated cortical activity that is governed by the damped wave equation with time-step $\delta t = 1$ and parameters $a = 1.5 \cdot 10^5$, $b = 2.5 \cdot 10^3$.
(TIF)

**S5 Fig. Resting-state fMRI and numerical simulation of the Wilson-Cowan graph neural field model on the human connectome.** Panel A shows resting-state brain activity, as fluctuations of the BOLD fMRI signal about the mean at each vertex. Panel B shows snapshots of activity from the stochastic Wilson-Cowan graph neural field model, simulated using the parameters of S2 Table. The model activity was temporally downsampled to match the TR of fMRI data, and rescaled by $\sqrt{\beta}$ to match the scale of the BOLD signal. No spatial or temporal smoothing was applied. Note that the two hemispheric surfaces are physically separate, and inter-hemispheric propagation is allowed through white matter fibers.
(TIF)

**S6 Fig. FMRI functional connectivity.** High-resolution PDF version of Fig 6.
(PDF)

**S7 Fig. Model functional connectivity.** High-resolution PDF version of Fig 7.
(PDF)

**S8 Fig. Functional connectivity comparison (inset 1).** High-resolution PDF version of Fig 8.
(PDF)

**S9 Fig. Functional connectivity comparison (inset 2).** High-resolution PDF version of Fig 9.
(PDF)

**S1 Appendix. Spatiotemporal convolutions on graphs.** Here, we generalize the formulation of spatial convolutions on graphs to spatiotemporal convolutions on graphs, allowing the definition of a broader class of graph neural fields.
(PDF)

**S2 Appendix. Reaction-diffusion neural activity models on graphs.** In this section we show how graph filters can also be used to implement the graph equivalents of neural activity models that can be directly written as partial differential equations [36, 53] and, among others, comprise damped wave and reaction-diffusion equations.
(PDF)

**S3 Appendix. Damped wave and telegrapher's equation on graphs.** The damped-wave describes the dynamics of simultaneous diffusion and wave propagation, and is thus of interest in the context of modelling activity propagation in neural tissue [53]. Nonlinear variants of the wave equation on graphs have also been the subject of previous analytical studies [67]. Here, we solve the graph equivalent of the damped-wave equation and of the telegrapher's equation,

which is of interest in the context of modelling action potentials [68].
(PDF)

**S4 Appendix. Wilson-Cowan model linear stability analysis.** In order to compute meaningful spatiotemporal observables with CHAOSS for a given set of parameters, it is first necessary to find a steady state and compute its stability to perturbations. Here, we provide solutions to the steady-state equations and a general linear stability analysis for the Wilson-Cowan model on graphs.
(PDF)

## Acknowledgments

We would like to thank Thomas Yeo and Ruby Kong for providing the mapping between HCP 32k and 10k vertices and Daniele Avitabile for valuable discussions.

## Author Contributions

**Conceptualization:** Marco Aqil, Selen Atasoy, Morten L. Kringelbach, Rikkert Hindriks.

**Formal analysis:** Marco Aqil, Rikkert Hindriks.

**Funding acquisition:** Morten L. Kringelbach, Rikkert Hindriks.

**Investigation:** Marco Aqil.

**Methodology:** Marco Aqil, Rikkert Hindriks.

**Project administration:** Marco Aqil, Selen Atasoy, Morten L. Kringelbach, Rikkert Hindriks.

**Resources:** Selen Atasoy, Morten L. Kringelbach.

**Software:** Marco Aqil.

**Supervision:** Selen Atasoy, Morten L. Kringelbach, Rikkert Hindriks.

**Visualization:** Marco Aqil.

**Writing – original draft:** Marco Aqil, Rikkert Hindriks.

**Writing – review & editing:** Marco Aqil, Selen Atasoy, Morten L. Kringelbach, Rikkert Hindriks.

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
