## [Decision Letter · Decision Letter 0]

7 Oct 2020

Dear Mr. Aqil,

Thank you very much for submitting your manuscript "Graph neural fields: a framework for spatiotemporal dynamical models on the human connectome" for consideration at PLOS Computational Biology.

As with all papers reviewed by the journal, your manuscript was reviewed by members of the editorial board and by several independent reviewers. In light of the reviews (below this email), we would like to invite the resubmission of a significantly-revised version that takes into account the reviewers' comments.

We cannot make any decision about publication until we have seen the revised manuscript and your response to the reviewers' comments. Your revised manuscript is also likely to be sent to reviewers for further evaluation.

Sincerely,

Daniele Marinazzo

Deputy Editor

PLOS Computational Biology

Reviewer's Responses to Questions

**Comments to the Authors:**

Reviewer #1: The paper presents an integration of concepts from graph signal processing and neural field models. In particular, the latter is extended to the former setting. To this reader, the work is original, innovative and has the potential to be leveraged to shed light in to the dynamics of resting state fMRI data through using local and distant backbone connections of the brain. The methodology, although description-wise not so nicely laid out, is rigorous and provides notable insights for the community. Despite its strengths, the paper has left this reader with two main concerns:

1) The paper is titled heavily in relation to the human connectome, whereas the contents are essentially theoretical derivation of general nature, which are applicable to any type of graph. Although the incorporation of the neural field models into the graph setting does put the work in context in regard to the neural setting, the majority of presented results are on a simplistic 1D graph of a non-neuronal nature. Results of the human connectome are limited to that of a single subject, the most substantial of which is to show that the power spectral density of a single time frame of the subject's resting-state fMRI data can be fitted with the proposed model. To this reader, this extent of results is extensively limited to prove the applicability and robustness of the proposed methodology on fMRI data, and as such, the work lacks substantial evidence for the conclusions made. A number of suggestions are presented in the comments that follow, which the authors may want to consider.

2) The presentation of the paper, in particular in relation to the mathematical formulations are not consistent and the associated descriptions are generally not concise. A series of errors and issues are listed in the comments that follow. To increase the readership of the work, the authors are highly recommended to consider thoroughly revising the manuscript for brevity.

Good luck.

a) In the paragraph before equation (33), you state: ‘the elements of M can be defined in terms of suitably scaled Euclidean distances, geodesic distances over the cortical manifold, or as the lengths of the white matter fibers connecting the vertices.’ It may benefit the reader if you could refer to related works for the listed scenarios, for instance, for the third one: https://doi.org/10.1109/TMI.2013.2271486, which defines intra-cortical graph edges based on the surface mesh in a similar way as done in your work. The work is also probably the first to present the idea designing hybrid graph with local and distant edges, and also uses non-binary edges, see equation (6).

b) As mentioned earlier above, results on a single subject is just too little. Can you extend results shown in Figure 7 across multiple subjects, potentially on the HCP100 data? If you normalize the fMRI graph signals to unit norm, and use the normalized graph Laplacian, which has its eigenvalues bounded within [0,2], you may then overlay the resulting power spectra, or the cumulative sum for instance as in Fig. 3(a) in: https://doi.org/10.1109/ISBI45749.2020.9098667, across subjects. Moreover, as your model is not specific to resting-state data, it can be intuitive to see the same results also on the HCP task data, at least for a subset of the subjects if computationally burdensome.

c) The FC matrices shown in Figure 5 are shown at a resolution that cannot be readily compared with each otehr. Could you please provide a zoomed inset of a region around the diagonal on both matrices, to enable pixel-by-pixel comparison of the two at the zoomed inset? Moreover, to better convey that the two matrices are very similar, you can show the difference between the two, and again, show a zoomed in region around the diagonal. Lastly, the colormap used does not nicely reflect positive-negative values with a nice separation of 0; in particular, you state: line 202: ‘The non-local edge also creates a visible increase in the functional connectivity between the nodes involved, and a change in the pattern in neighboring nodes’ which is not at all as visible as you state. If you modify your colormap, for instance to show 0 as white, things should become much more visible. In doing so, you may want to use the following in matlab: RdBu = flipud(cbrewer('div','RdBu',101)); colormap(RdBu); for which you need to download and addpath this package: https://www.mathworks.com/matlabcentral/fileexchange/34087-cbrewer-colorbrewer-schemes-for-matlab

d) On line 526 you state: ‘intra-cortical edges are weighed by they 3D Euclidean distance’. Can you more specifically state how you define these weights? In particular, do you divide or multiply by the Euclidean distance? (Also note the typo: they > their.) But aside from this, I found your approach in scaling of the non-local edge weights very intuitive and an interesting way to account for intra-cortical propagation.

e) On line 607 you state: ‘Graph neural fields can naturally take into account important physical properties such as cortical folding, hemispheric asymmetries, non-homogeneous structural connectivity, and white matter projections, with a minimal amount of computational power.’ This is indeed interesting, but can you please elaborate a bit on how this is achievable?

f) The connectome model, when used for the sake of representation as the backbone on which fMRI data are realized, should be interpreted with care, in particular, in light of, firstly, limitations with tractography in regards to false positive and true negatives, and secondly, lack of projection fibers in the model that connect the cortex and subcortical structures such as the thalamus, basal ganglia, and spinal cord and the cerebellum. The concern is not only lack of connections, but in fact lack of vertices representing these structures. I understand that the presented work is not expected to address these concerns, but I believe these concerns deserve be mentioned, at least briefly, in the Discussion, to inform the reader.

g) The notation P(k) used in the y-axis label of Figure 7 seems not to appear in text, but you have H_{p}(k), though I do not perfectly follow what would be the 'neuronal population' in this example figure if relevant.

h) On line 594 you state that: ‘Whole-brain models that incorporate short-range connectivity are referred to as surface- based because they can are defined either on high-resolution surface-based representations of the cortex’. However, models that incorporate short-range connectivity can also be volumetric, e.g. see: [whole brain graph] https://doi.org/10.1016/j.neuroimage.2020.116718, [white matter graph] https://doi.org/10.1109/ISBI45749.2020.9098582, [gray matter graph] https://doi.org/10.1016/j.neuroimage.2015.06.010. There is no need to refer to these works as their aim differs in nature from that in your work, aside from the first one to some extent. The concern is with the sentence that is not quite accurate.

i) At the C Elegans statement, line 668, you may find the following work of relevance to refer to: https://doi.org/10.1162/netn_a_00084

j) To give the reader a more recent comprehensive review of GSP and its applications consider citing the following: https://doi.org/10.1016/j.dsp.2020.102802

k) You may want to state that the filtering operation of graphs using any desired spectral kernel can be efficiently implemented using polynomial approximation scheme which obviates the need to diagonalize the Laplacian matrices, which in your setting of 18K nodes, although feasible, but still quite computationally cumbersome to get all the eigenmodes. If you decided to mention such an option as outlook, you may find this recent preprint of interest to refer to as it provides a nice overview of available techniques: https://arxiv.org/abs/2006.11220

l) In the paragraph above equation (75), it states: ‘Because of the independence of eigenmodes,…’. I suppose you mean 'orthogonality' of the eigenmodes? (Also note that, i) the type at the end of the paragraph: spectrum. > spectrum: --- ii) sometimes you use: when presenting equations and sometimes not; please consider making the presentation consistent)

m) For constants, you sometimes use lower case letters, like n that specifies the number of eigenmodes, and sometimes use uppercase, like N that specifies the number of neuronal populations.

n) You redefine Dt multiple times and in an inconsistent way. Firstly, this is a very inconcise, and disturbing for the reader. Secondly, you sometimes refer to it as: 'where Dt is A temporal differential operator', and sometimes as: 'where Dt is THE temporal differential operator'. Moreover, you apparently use three different notations for temporal differentiation, which is quite sloppy: i) \\dot{.}, ii) \\frac{\\partial .}{\\partial t}, iii) Dt .

o) In describing graphs, sometimes you use the term ‘nodes’ and sometimes ‘vertices’. Please stick with one.

p) Table 1 should either be placed earlier, or be referred to early on in the section to prevent losing the reader.

q) It would be much better to present the diagonalization of the Laplacian and related notations (lines 166-117) before/after eq. (2); then you can skip the related descriptions of the degree matrix, after eq (4), and U matrix in eqs. (11)-(12). You are also repeating the diagonalization relation before eq. (38). Please do thoroughly re-read your paper, throughout, as there are many such instances of unnecessary and redundant repetitions.

r) You use multiple notations for vectors: bold lower case, upper case, bold upper case, for instance ‘d’, ‘K’ and ‘X’ in eq. (13). This also happens in the other equations. This is sloppy and super confusing for the reader.

s) The sentence after eq. (17) is either incomplete or badly written if it is linked to (17).

t) In the description after eq. (18), you denote S as S(.) whereas in the definition you use S[.].

u) Sometimes you say 'eigenmode k' and sometimes 'kth eigenmode'. Please consider making the description consistent and stick to one scheme.

v) Some other typos:

- After equation (73): Where > where

- After equation (74): where 1/2\\pi is. > this is obvious, so you may drop it.

- There are many repetitive sentences. For instance: ‘where n/N is…’. It should be sufficient to define these once.

- Line 106: ‘…5s of…’ > ‘…5 seconds of…’

- Line 515: (i; j)^{th} > is th needed?

- Line 579: varies > various

- Comma missing at the end of equations (10), (24), (67), ?

- Full stop missing at the end of equation (23), (17), (22), (20), (18), (19), (40), ?

- After eqs. (4), (10), etc. start sentence with ‘where’; e.g. ‘where’ \\Delta is..

- Before eq (11): are matrices of size (n; n) > are $n \\times n$ matrices.

- Page 8, second line: solutions the steady… > solutions TO the steady...

- Page 9, first sentence: abut > about

Reviewer #2: This paper introduces a graph neural field approach to describe whole brain neural activity which allows to derive spectra and also functional connecitivity. The authors use concepts from neural fields and graph signal processing to achieve this. Overall, I am very enthusiastic about this work and it definitely contributes to the growing interest and role of the eigenmodes to brain activity and connectivity. The math is rigorous and can be clearly followed. I do have a few concerns.

Major concerns

My suggestion would be to reorder and shorten the paper. The paper in its current form is quite long. The current ordering of the sections is also confusing and not according to the journal’s guidelines. I would suggest to have the following order: introduction, results, discussion, methods. I think there are parts of the paper that were necessary as initial sanity checks, such as “damped wave equation on the human connectome” in the result section and “Reaction-diffusion neural activity models”. Though relevant for following the reasoning and steps in the paper, these sections also quite distract from the main storyline of the paper. I understand the role of the linear stability analysis applied to the Wilson-Cowan equations. However, this type of analysis on Wilson-Cowan equations on a network is not new (see (Tewarie et al., 2019)). I would suggest to put all these/or part of these sections to the supplemental material.

The authors use a Gaussian Kernel for their Wilson-Cowan model based analysis. The authors do illustrate the possibilities of the kernels (see method section). Maybe I haven’t noticed it, but I haven’t seen any analysis on the influence of these different kernels. The choice of the kernel can actually have dramatic influence on the stability of the steady states and the model’s bifurcation structure (Atay and Hutt, 2004; Hutt and Atay, 2005; Shamsara et al., 2019). The authors have merely used the Gaussian Kernel with one value for the standard deviation. It is not clear to me how this standard deviation was the result of some optimization. Nor is it clear to me whether this Kernel also optimized the resemblance of functional connectivity with empirical data. Why did the authors not use a range of sigma? It would be interesting if the authors can discuss or speculate on the usage of different kernels for E and I cells, for some I cells, there is evidence of lateral inhibition, which would advocate for example for a Mexican hat kernel for I population (in the Wilson-Cowan example).

It is not clear to me how well the prediction of the model is for empirical functional connectivity patterns from fMRI. The authors do show how well the modeled spectra match the empirical one in Figure 7. However, equally important is whether modeled activity patterns or functional connectivity patterns match the empirical data. I doubt whether functional connectivity patterns such as in figure 6 match empirical fMRI based connectivity matrices.

I cannot find any mentioning of distance dependent delays. Can the authors discuss or mention if distance dependence delays for the propagation along white matter tracts can be implemented in their approach ((Alswaihli et al., 2018)), and otherwise comment how their distance weighting would make this less important? Could the authors please clarify how the distance was set for the Wilson-cowan based simulations on page 11. Did the authors use the distance along the cortical surface for local edges? Distance along the white matter tract for long range connections? If not, I would suggest the authors to implement this. I do not understand on page 11 that the non-local connections are merely based on DTI data? Cortico-cortical connections do not necessarily go through white matter tracts, but can propagate through connections between infra/supragranular layers in the cortex.

Minor concerns

I would put “eigenmodes” in the keywords, or even in the title.

Are sigma in equations (9) and (10) the same as sigma in (11) and (12), i.e. to they correspond to the standard deviation of Gaussian Kernels.

Maybe I overlooked this, but please explicitly mention what the definition of d_j is in (33).

The authors could consider to rename some state variables. The matrix U contains the eigenvectors of the graph Laplacian. At the same time there is state variable u(t). When people scan your paper, they can confuse u to be columns of U. I had this confusion too when I scanned the paper at first. I know that u(x,t) is often used in the field of neural fields to describe the state variables, maybe you could consider to rename the eigenvectors of the graph Laplacian?

I assume equations (62), (65), (67), (69) refer to type 2 neural field equations. Please explicitly mention to make it easy for the reader.

I think this statement in the discussion needs references: “This prevents studying the mechanisms underlying a large class of cortical activity patterns that have been observed in experiments, including traveling and spiral waves, sink-source-dynamics as well as their role in shaping macroscopic dynamics.”

There is a spelling error in the first sentence of the 4th paragraph in the discussion:” based because they can are defined either on high-resolution”. This is either “are” or “can”.

I would suggest to use consistent terminology, it is either nodes and links (networks) or vertices and edges (graphs). The authors use edges and links both at the same time with vertices.

Some recent literature deserves to be mentioned in the context of eigenmodes expression in functional MRI or EEG data, see recent papers: (Glomb et al., 2020; Preti and Van De Ville, 2019; Tewarie et al., 2020). How much different is the damped wave equation on the connectome compared to (Caputo et al., 2017).

Alswaihli, J., Potthast, R., Bojak, I., Saddy, D., and Hutt, A. (2018). Kernel reconstruction for delayed neural field equations. J. Math. Neurosci. 8, 3.

Atay, F. M., and Hutt, A. (2004). Stability and bifurcations in neural fields with finite propagation speed and general connectivity. SIAM J. Appl. Math. 65, 644–666.

Caputo, J.-G., Khames, I., Knippel, A., and Panayotaros, P. (2017). Periodic orbits in nonlinear wave equations on networks. J. Phys. A Math. Theor. 50, 375101.

Glomb, K., Queralt, J. R., Pascucci, D., Defferrard, M., Tourbier, S., Carboni, M., et al. (2020). Connectome spectral analysis to track EEG task dynamics on a subsecond scale. Neuroimage 221, 117137.

Hutt, A., and Atay, F. M. (2005). Analysis of nonlocal neural fields for both general and gamma-distributed connectivities. Phys. D Nonlinear Phenom. 203, 30–54.

Preti, M. G., and Van De Ville, D. (2019). Decoupling of brain function from structure reveals regional behavioral specialization in humans. Nat. Commun. 10, 1–7.

Shamsara, E., Yamakou, M. E., Atay, F. M., and Jost, J. (2019). Dynamics of neural fields with exponential temporal kernel. arXiv Prepr. arXiv1908.06324.

Tewarie, P., Abeysuriya, R., Byrne, Á., O’Neill, G. C., Sotiropoulos, S. N., Brookes, M. J., et al. (2019). How do spatially distinct frequency specific MEG networks emerge from one underlying structural connectome? The role of the structural eigenmodes. Neuroimage 186, 211–220.

Tewarie, P., Prasse, B., Meier, J. M., Santos, F. A. N., Douw, L., Schoonheim, M., et al. (2020). Mapping functional brain networks from the structural connectome: relating the series expansion and eigenmode approaches. Neuroimage 216, 116805. doi:https://doi.org/10.1016/j.neuroimage.2020.116805.

Reviewer #3: The review has been uploaded as an attachment

**Have all data underlying the figures and results presented in the manuscript been provided?**

Reviewer #1: Yes

Reviewer #2: Yes

Reviewer #3: Yes

PLOS authors have the option to publish the peer review history of their article (what does this mean?). If published, this will include your full peer review and any attached files.

Reviewer #1: **Yes: **Hamid Behjat

Reviewer #2: No

Reviewer #3: **Yes: **Enrico Cataldo
---

## [Decision Letter · Decision Letter 1]

11 Dec 2020

Dear Mr. Aqil,

We are pleased to inform you that your manuscript 'Graph neural fields: a framework for spatiotemporal dynamical models on the human connectome' has been provisionally accepted for publication in PLOS Computational Biology.

Best regards,

Daniele Marinazzo

Deputy Editor

PLOS Computational Biology

Daniele Marinazzo

Deputy Editor

PLOS Computational Biology

Reviewer's Responses to Questions

**Comments to the Authors:**

Reviewer #1: This reviewer's previous comments have been sufficiently addressed/responded to. Thank you for investing considerable effort in improving the manuscript, through enhancing the organization and adding complementary results.

Reviewer #2: I very much appreciate the effort of the authors. They have sufficiently addressed all my comments and I am happy to recommend this paper for publication.

Reviewer #3: Dear Editor,

the authors have updated the paper, adding the suggested changes.

The paper can be accepted for publication.

**Have all data underlying the figures and results presented in the manuscript been provided?**

Reviewer #1: Yes

Reviewer #2: None

Reviewer #3: Yes

PLOS authors have the option to publish the peer review history of their article (what does this mean?). If published, this will include your full peer review and any attached files.

Reviewer #1: **Yes: **Hamid Behjat

Reviewer #2: No

Reviewer #3: No

---

## [Editor Report · Acceptance letter]

23 Jan 2021

PCOMPBIOL-D-20-01590R1 

Graph neural fields: a framework for spatiotemporal dynamical models on the human connectome

Dear Dr Aqil,

I am pleased to inform you that your manuscript has been formally accepted for publication in PLOS Computational Biology. Your manuscript is now with our production department and you will be notified of the publication date in due course.

With kind regards,

Melanie Wincott
